# SVFT: Parameter-Efficient Fine-Tuning with Singular Vectors

**Vijay Lingam**[† §*]    **Atula Tejaswi**[†*]    **Aditya Vavre**[†*]    **Aneesh Shetty**[†*]

**Gautham Krishna Gudur**[†*]    **Joydeep Ghosh**[†]    **Alex Dimakis**[†]    **Eunsol Choi**[†]

**Aleksandar Bojchevski**[‡*]    **Sujay Sanghavi**[†*]

[†]University of Texas at Austin    [‡]University of Cologne

[§]CISPA Helmholtz Center for Information Security

## Abstract

Popular parameter-efficient fine-tuning (PEFT) methods, such as LoRA and its variants, freeze pre-trained model weights $\mathbf{W}$ and inject learnable matrices $\mathbf{\Delta W}$. These $\mathbf{\Delta W}$ matrices are structured for efficient parameterization, often using techniques like low-rank approximations or scaling vectors. However, these methods typically exhibit a performance gap compared to full fine-tuning. While recent PEFT methods have narrowed this gap, they do so at the expense of additional learnable parameters. We propose SVFT[2], a *simple* approach that structures $\mathbf{\Delta W}$ based on the specific weight matrix $\mathbf{W}$. SVFT updates $\mathbf{W}$ as a sparse combination $M$ of outer products of its singular vectors, training only the coefficients of these combinations. Crucially, we make additional off-diagonal elements in $M$ learnable, enabling a smooth trade-off between trainable parameters and expressivity—an aspect that distinctly sets our approach apart from previous works leveraging singular values. Extensive experiments on language and vision benchmarks show that SVFT recovers up to **96%** of full fine-tuning performance while training only **0.006 to 0.25**% of parameters, outperforming existing methods that achieve only up to **85%** performance with **0.03 to 0.8**% of the trainable parameter budget.

## 1 Introduction

Large-scale foundation models are often adapted for specific downstream tasks after pre-training. Parameter-efficient fine-tuning (PEFT) facilitates this adaptation efficiently by learning a minimal set of new parameters, thus creating an "expert" model. For instance, Large Language Models (LLMs) pre-trained on vast training corpora are fine-tuned for specialized tasks such as text summarization [13, 37], sentiment analysis [27, 21], and code completion [28] using instruction fine-tuning datasets. Although full fine-tuning (Full-FT) is a viable method to achieve this, it requires re-training and storing all model weights, making it impractical for deployment with large foundation models.

To address these challenges, PEFT techniques [14] (e.g., LoRA [15]) were introduced to significantly reduce the number of learnable parameters compared to Full-FT, though often at the cost of performance. DoRA [19] bridges this performance gap by adding more learnable parameters and being more expressive than LoRA. Almost all these methods apply a low-rank update additively to the frozen pre-trained weights, potentially limiting their expressivity. Furthermore, these adapters are agnostic to the structure and geometry of the weight matrices they modify. Finally, more expressive

---

[*]indicates equal contribution/advising

[2]Code is available at `https://github.com/VijayLingam95/SVFT/`

38th Conference on Neural Information Processing Systems (NeurIPS 2024).

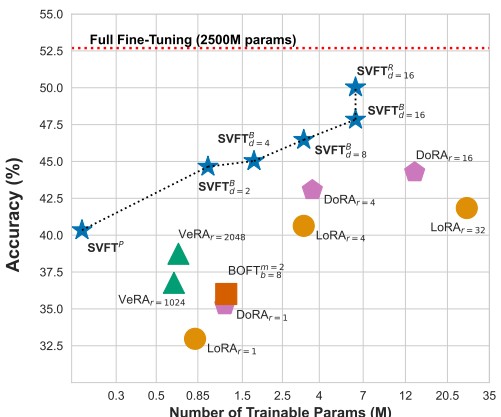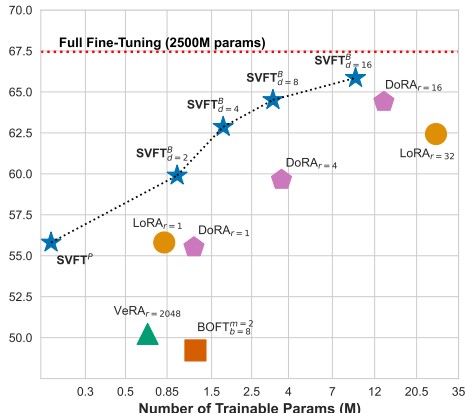

Figure 1: Performance vs total trainable parameters for GSM-8K (left) and Commonsense Reasoning (right) on Gemma-2B. $\text{SVFT}_{d=16}^{B/R}$ outperforms $\text{DoRA}_{r=8/16}$ with 75% less trainable parameters.

PEFT methods (e.g., LoRA, DoRA, BOFT [20]) still accumulate a considerable portion of learnable parameters even in their most efficient configuration (e.g., setting rank=1 in LoRA and DoRA). The storage requirements for the learnable adapters can grow very quickly when adapting to a large number of downstream tasks [17].

Is it possible to narrow the performance gap between PEFT and Full-FT while being highly parameter-efficient? Yes, we propose SVFT: Singular Vectors guided Fine-Tuning — a *simple* approach that involves updating an existing weight matrix by adding to it a sparse weighted combination of *its own singular vectors*. The structure of the induced perturbation in SVFT depends on the specific matrix being perturbed. Our contributions can be summarized as follows:

- We introduce SVFT, a new PEFT method. Given a weight matrix $W$, SVFT involves adapting it with a matrix $\Delta W := \sum_{(i,j) \in \Omega} m_{ij} u_i v_j^T$ where the $\{u_i\}$ and $\{v_j\}$ are the left and right singular vectors of $W$, $\Omega$ is an a-priori fixed sparsity pattern, and $m_{ij}$ for $(i,j) \in \Omega$ are learnable parameters. By controlling $|\Omega|$ we can efficiently explore the accuracy vs parameters trade-off.

- SVFT achieves higher downstream accuracy, as a function of the number of trainable parameters, as compared to several popular PEFT methods (see Figure 1) and over several downstream tasks across both vision and language tasks. For instance, on GSM-8K using Gemma-2B our method recovers up to **96%** of full fine-tuning performance while training only **0.006 to 0.25%** of parameters, outperforming existing methods that only recover up to **85%** performance using **0.03 to 0.8%** the trainable parameter budget (see Figure 1).

We introduce four simple variants for parameterizing weight updates, namely: *Plain*, *Random*, *Banded*, and *Top-k* in SVFT (which differ in their choices of the fixed sparsity pattern $\Omega$) and validate these design choices empirically. Additionally, we theoretically show that for any fixed parameters budget, SVFT can induce a higher rank perturbation compared to previous PEFT techniques.

## 2 Related Work

Recent advancements in large language models (LLMs) have emphasized the development of PEFT techniques to enhance the adaptability and efficiency of large pre-trained language models.

**LoRA.** A notable contribution in this field is Low-Rank Adaptation (LoRA) [15], which freezes the weights of pre-trained models and integrates trainable low-rank matrices into each transformer layer. For a pre-trained weight matrix $W_0 \in \mathbb{R}^{d \times n}$, LoRA constraints the weight update $\Delta W$ to a low-rank decomposition: $h = W_0 x + \Delta W x = W_0 x + \underline{BA} x$, where $B \in \mathbb{R}^{d \times r}$, $A \in \mathbb{R}^{r \times n}$ and rank $r \ll \min(d, n)$. We underline the (trainable) parameters that are updated via gradient descent.

**LoRA variants.** We highlight some recent approaches that further improve the vanilla LoRA architecture. Vector-based Random Matrix Adaptation (VeRA) [17] minimizes the number of

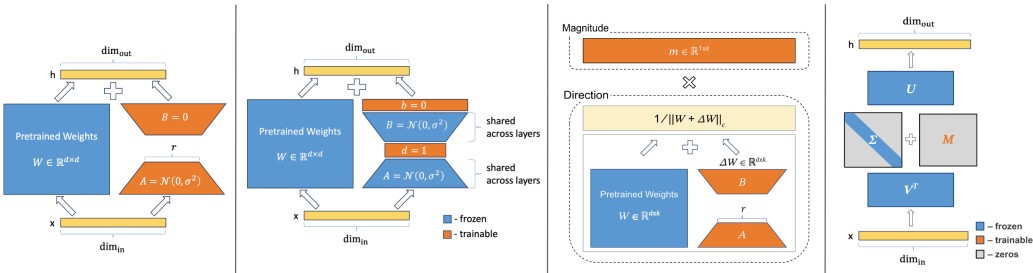

Figure 2: Schematic comparison of LoRA, VeRA, DoRA, and SVFT (left to right).

trainable parameters by utilizing a pair of low-rank random matrices shared between layers and learning compact scaling vectors while maintaining performance comparable to LoRA. Formally, VeRA can be expressed as: $\boldsymbol{h} = \boldsymbol{W}_0\boldsymbol{x} + \Delta\boldsymbol{W}\boldsymbol{x} = \boldsymbol{W}_0\boldsymbol{x} + \underline{\boldsymbol{\Lambda}_b}\boldsymbol{B}\underline{\boldsymbol{\Lambda}_d}\boldsymbol{A}\boldsymbol{x}$, where $\boldsymbol{A}$ and $\boldsymbol{B}$ are initialized randomly, frozen, and shared across layers, while $\boldsymbol{\Lambda}_b$ and $\boldsymbol{\Lambda}_d$ are trainable diagonal matrices.

An alternative approach, Weight-Decomposed Low-Rank Adaptation (DoRA) [19], decomposes pre-trained weight matrices into magnitude and direction components, and applies low-rank updates for directional updates, reducing trainable parameters and enhancing learning capacity and training stability. DoRA can be expressed as: $\boldsymbol{h} = \underline{\boldsymbol{m}}\frac{\boldsymbol{W}_0 + \Delta\boldsymbol{W}}{\|\boldsymbol{W}_0 + \Delta\boldsymbol{W}\|_c}\boldsymbol{x} = \underline{\boldsymbol{m}}\frac{\boldsymbol{W}_0 + \underline{\boldsymbol{B}\boldsymbol{A}}}{\|\boldsymbol{W}_0 + \underline{\boldsymbol{B}\boldsymbol{A}}\|_c}\boldsymbol{x}$, where $\|\cdot\|_c$ denotes the vector-wise norm of a matrix across each column. Similar to LoRA, $\overline{\boldsymbol{W}}_0$ remains frozen, whereas the magnitude vector $\boldsymbol{m}$ (initialized to $\|\boldsymbol{W}_0\|_c$) and low-rank matrices $\boldsymbol{A}, \boldsymbol{B}$ contain trainable parameters.

AdaLoRA [38] adaptively distributes the parameter budget across weight matrices based on their importance scores and modulates the rank of incremental matrices to manage this allocation effectively. PiSSA (Principal Singular Values and Singular Vectors Adaptation) [22] is another variant of LoRA, where matrices $\boldsymbol{A}, \boldsymbol{B}$ are initialized with principal components of SVD and the remaining components are used to initialize $\boldsymbol{W}_0$. FLoRA [34] enhances LoRA by enabling each example in a mini-batch to utilize distinct low-rank weights, preserving expressive power and facilitating efficient batching, thereby extending the domain adaptation benefits of LoRA without batching limitations.

**Other PEFT variants.** Orthogonal Fine-tuning (OFT) [26] modifies pre-trained weight matrices through orthogonal reparameterization to preserve essential information. However, it still requires a considerable number of trainable parameters due to the high dimensionality of these matrices. Butterfly Orthogonal Fine-tuning (BOFT) [20] extends OFT's methodology by incorporating Butterfly factorization thereby positioning OFT as a special case of BOFT. Unlike the additive low-rank weight updates utilized in LoRA, BOFT applies multiplicative orthogonal weight updates, marking a significant divergence in the approach but claims to improve parameter efficiency and fine-tuning flexibility. BOFT can be formally expressed as: $\boldsymbol{h} = (\underline{\boldsymbol{R}(m, b)} \cdot \boldsymbol{W}_0)\boldsymbol{x}$, where the orthogonal matrix $\boldsymbol{R}(m, b) \in \mathbb{R}^{d \times d}$ is composed of a product of multiple orthogonal butterfly components. When $m = 1$, BOFT reduces to block-diagonal OFT with block size $b$. When $m = 1$ and $b = d$, BOFT reduces to the original OFT with an unconstrained full orthogonal matrix.

**SVD-based Variants.** SVF [31], SVDiff [10], and SAM-Parser [25] also leverage the structure of $W$ matrices by decomposing them into three consecutive matrices via Singular Value Decomposition (SVD). However, these methods fine-tune only the singular values while keeping other components fixed, making them comparable to $\text{SVFT}^P$. In Appendix C.1, we present a comparison of $\text{SVFT}^P$ with SVF, confirming that their performance is similar, which supports our observations.

## 3 Method

In this section, we introduce Singular Vectors guided Fine-Tuning (SVFT). The main innovation in SVFT lies in applying structure/geometry-aware weight updates through sparse weighted combination of singular vectors.

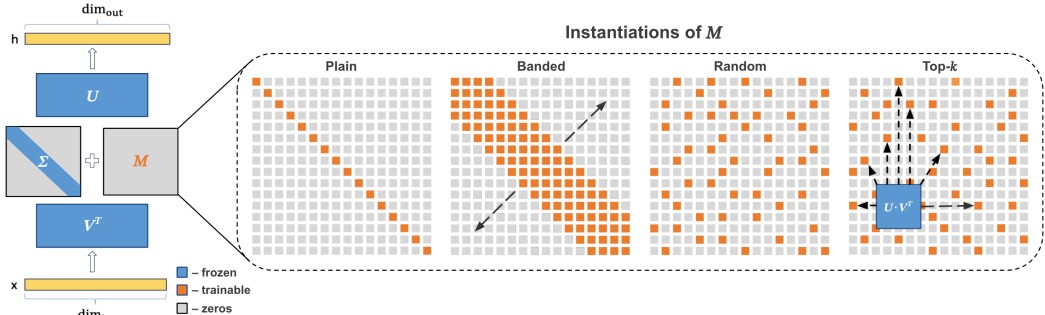

Figure 3: An Overview of SVFT. The original weights $\boldsymbol{W}$ are decomposed into $\boldsymbol{U}, \boldsymbol{\Sigma}, \boldsymbol{V}$. Here, $\boldsymbol{M}$ contains all the trainable parameters, which can be configured into patterns such as Plain, Random, Banded, and Top-$k$, represented by patterns of trainable (orange) and zero (gray) elements.

## 3.1 SVFT Formulation

We now formally describe our method, SVFT for parameter-efficient fine-tuning of a pre-trained model. Let $\boldsymbol{W}_0 \in \mathbb{R}^{d_1 \times d_2}$ denote a weight matrix in the pre-trained model, such as a key matrix, query matrix, or an MLP matrix within a transformer block. To this matrix, we add a structured, learnable update $\Delta \boldsymbol{W}$ as follows.

As a first step, we compute the Singular Value Decomposition (SVD) of the given matrix: $\boldsymbol{W}_0 = \boldsymbol{U} \boldsymbol{\Sigma} \boldsymbol{V}^T$. That is, $\boldsymbol{U}$ is the $d_1 \times d_1$ matrix of left singular vectors (i.e., its columns are orthonormal), $\boldsymbol{V}^T$ is the $d_2 \times d_2$ matrix of right singular vectors (i.e., its rows are orthonormal), and $\boldsymbol{\Sigma}$ is a $d_1 \times d_2$ diagonal matrix. Then, we parameterize our weight update as $\Delta \boldsymbol{W} = \boldsymbol{U} \underline{\boldsymbol{M}} \boldsymbol{V}^T$, where $\boldsymbol{U}, \boldsymbol{V}$ are fixed and frozen, while $\underline{\boldsymbol{M}}$ is a $d_1 \times d_2$ **sparse trainable matrix with pre-determined and fixed sparsity pattern**[3]. That is, we first pre-determine a small fixed set of elements in $\boldsymbol{M}$ that will be allowed to be non-zero and train only those elements. The forward pass for SVFT can be written as,

$$h = \boldsymbol{W}_0 x + \Delta \boldsymbol{W} x = \boldsymbol{U}(\boldsymbol{\Sigma} + \underline{\boldsymbol{M}}) \boldsymbol{V}^T \boldsymbol{x} \tag{1}$$

We explore four simple choices for $\Omega$, the pre-determined sparsity pattern of $\underline{\boldsymbol{M}}$.

**Plain** $\left(\text{SVFT}^P\right)$. In this variant, we constrain $\underline{\boldsymbol{M}}$ to be a diagonal matrix, which can be interpreted as adapting singular values and reweighting the frozen singular vectors. Since only the diagonal elements are learned, this is the most parameter-efficient SVFT variant.

**Banded** $\left(\text{SVFT}^B_d\right)$. In this approach, we populate $\underline{\boldsymbol{M}}$ using a banded matrix, progressively making off-diagonals learnable. Specifically, for constants $z_1$ and $z_2$, $\boldsymbol{M}_{ij} = 0$ if $j < i - z_1$ or $j > i + z_2$, where $z_1, z_2 \geq 0$. In our experiments, we set $z_1 = z_2 = d$ to induce off-diagonal elements that capture additional interactions beyond those represented by singular values. This banded perturbation induces local interactions, allowing specific singular values to interact with their immediate neighbors, ensuring smoother transitions. This method, although deviating from the canonical form of SVD, provides a mechanism to capture localized interactions.

**Random** $\left(\text{SVFT}^R_d\right)$. A straightforward heuristic for populating $\underline{\boldsymbol{M}}$ involves randomly selecting $k$ elements to be learnable.

**Top-**$k$ $\left(\text{SVFT}^T_{\#p}\right)$. The final design choice we explore involves computing the alignment between the left and right singular vectors as $\boldsymbol{u}_i^T \boldsymbol{v}_j$. We then select the top-$k$ elements and make them learnable. However, note that this only works when left and right singular vectors have the same size. A possible interpretation of this is we make only the top-$k$ strong interactions between singular vector directions learnable. The subscript $\#p$ denotes the total number of learnable parameters.

We illustrate these SVFT design choices in Figure 3. Our empirical results demonstrate that these simple design choices significantly enhance performance compared to state-of-the-art PEFT methods. Note that $\text{SVFT}^P$ has a fixed number of learnable parameters, while the remaining variants are configurable. We hypothesize that further innovation is likely achievable through optimizing the sparsity pattern of $\underline{\boldsymbol{M}}$, including efficient learned-sparsity methods. In this paper, we explore these

---

[3]Learnable parameters are underlined.

four choices to validate the overall idea: determining a perturbation using the singular vectors of the matrix that is being perturbed.

## 3.2 Properties of SVFT

We highlight some properties of SVFT in the following lemma and provide insights into how its specific algebraic structure compares and contrasts with baseline PEFT methods.

**Lemma:** Let $W_0$ be a matrix of size $d_1 \times d_2$ with SVD given by $U \Sigma V^T$. Consider an updated final matrix $W_0 + U M V^T$, where $M$ is a matrix of the same size as $\Sigma$, which may or may not be diagonal. Then, the following holds:

  (a) *Structure:* If $M$ is also diagonal (i.e. the plain SVFT), then the final matrix $W_0 + U M V^T$ has $U$ as its left singular vectors and $\mathrm{sign}(\Sigma + M) V^T$ as its right singular vectors. That is, its singular vectors are unchanged, except for possible sign flips. Conversely, if $M$ is *not* diagonal (i.e., variants of SVFT other than plain), then $U$ and $V$ may no longer be the singular directions of the final matrix $W_0 + U M V^T$.

  (b) *Expressivity:* Given *any* target matrix $P$ of size $d_1 \times d_2$, there exists an $M$ such that $P = W_0 + U M V^T$. That is, if $M$ is fully trainable, any target matrix can be realized using this method.

  (c) *Rank:* If $M$ has $k$ non-zero elements, then the rank of the update $U M V^T$ is at most $\min\{k, \min\{d_1, d_2\}\}$. For the same number of trainable parameters, SVFT can produce a much higher rank perturbation than LoRA (eventually becoming full rank), but in a constrained structured subspace.

We provide our proofs in Appendix A. Building on this lemma, we now compare the form of the SVFT update with LoRA and VeRA. SVFT's $\Delta W$ can be written as a sum of rank-one matrices:

$$\Delta W = \sum_{(i,j) \in \Omega} \underline{m_{ij}} \boldsymbol{u}_i \boldsymbol{v}_j^T \tag{2}$$

where $\boldsymbol{u}_i$ is the $i^{th}$ left singular vector, $\boldsymbol{v}_j$ is the $j^{th}$ right singular vector, and $\Omega$ is the set of non-zero elements in $M$. Thus, our method involves adding a weighted combination of specific rank-one perturbations of the form $\boldsymbol{u}_i \boldsymbol{v}_j^T$.

LoRA and VeRA updates can also be expressed as sums of rank-one matrices.

$$\Delta W_{\text{LoRA}} = \sum_{i=1}^{r} \underline{\boldsymbol{a}_i} \, \underline{\boldsymbol{b}_i}^T \quad \text{and} \quad \Delta W_{\text{VeRA}} = \sum_{i=1}^{r} \underline{\alpha_i} (\hat{\boldsymbol{a}}_i \odot \underline{\boldsymbol{\beta}}) \hat{\boldsymbol{b}}_i^T \tag{3}$$

where $\underline{\boldsymbol{a}_i}$ and $\boldsymbol{b}_j$ are the trainable columns of $A$ and $B$ matrices in LoRA. In VeRA, $\hat{\boldsymbol{a}}_i$ and $\hat{\boldsymbol{b}}_i$ are random and fixed vectors, while $\underline{\boldsymbol{\alpha}}$ and $\underline{\boldsymbol{\beta}}$ represent the diagonal elements of $\Lambda_d$ and $\Lambda_b$ respectively.

Note that LoRA requires $d_1 + d_2$ trainable parameters per rank-one matrix, while SVFT and VeRA require only one. Although LoRA can potentially capture directions different from those achievable by the fixed $\{\boldsymbol{u}_i, \boldsymbol{v}_j^T\}$ pairs, each of these directions incurs a significantly higher parameter cost.

VeRA captures new directions at a parameter cost similar to SVFT; however, there is a key distinction: in VeRA, each vector $\hat{\boldsymbol{a}}_i$ or $\hat{\boldsymbol{b}}_i$ appears in only one of the rank-one matrices. In contrast, in SVFT, the same vector $\boldsymbol{u}_i$ can appear in multiple terms in the summation, depending on the sparsity pattern of $M$. This results in an important difference: unlike SVFT, VeRA is *not universally expressive* – it cannot represent any target matrix $P$. Moreover, $\hat{\boldsymbol{a}}_i, \hat{\boldsymbol{b}}_i$ are random, while $\boldsymbol{u}_i, \boldsymbol{v}_j$ depend on $W_0$.

**Note.** SVFT requires storing both left and right singular vectors due to its computation of the SVD on pre-trained weights. While this increases memory usage compared to LoRA, it remains comparable to or lower than DoRA and BOFT. We present a memory analysis in Section 5.3. Further exploration of memory-reduction techniques, such as quantization, is planned as future work. Importantly, inference time and memory consumption remain the same across all methods, including SVFT, as the weights can be fused.

# 4 Experiments

## 4.1 Base Models & Setup

We adapt widely-used language models, encoder-only model (DeBERTaV3$_{base}$ [11]) and two decoder-only models (Gemma-2B/7B [32], LLaMA-3-8B [1]). We also experiment with vision transformer models (ViT-B/16 and ViT-L/16) [9]) pre-trained on ImageNet-21k [8], following prior work [17]. The complete details of our experimental setup and hyperparameter configurations are provided in Appendix C.

**Baselines.** We compare with **Full Fine-Tuning (FT)** updating all learnable parameters in all layers, along with **LoRA** [15], **DoRA** [19], **BOFT** [20] and **VeRA** [17].[4]

**Target Modules.** We adapt *all weight matrices* for SVFT, as it does not increase trainable parameters at the same rate as baseline methods. For baselines, we adapt the target modules recommended in [19]: QKVUD matrices for LoRA and DoRA, compatible matrices for VeRA, and QV matrices for BOFT to stay within GPU memory limits. Additional details can be found in Appendix C.7 and C.8. We also conduct experiments adapting QKVUD modules across methods and observe similar trends, as discussed in Appendix C.2.

## 4.2 Datasets

**Language.** For natural language generation (NLG) tasks, we evaluate on GSM-8K [7] and MATH [12] by fine-tuning on MetaMathQA-40K [35], following [20]. We also evaluate on 8 commonsense reasoning benchmarks (BoolQ [5], PIQA [3], SIQA [30], HellaSwag [36], Winogrande [29], ARC-easy/challenge [6], and OpenBookQA [23]). We follow the setting outlined in prior work [19, 16], where the training sets of all benchmarks are amalgamated for fine-tuning. We fine-tune on 15K examples from this training set. For natural language understanding (NLU), we evaluate on the General Language Understanding Evaluation (GLUE) benchmark consisting of classification and regression tasks, in line with [17, 15].

**Vision.** Our experiments on vision tasks consist of 4 benchmarks: CIFAR-100 [18], Food101 [4], RESISC45 [33], and Flowers102 [24]. We follow the setup from [17], and fine-tune on a subset comprising 10 samples from each class.

Table 1: Performance (Accuracy) on Mathematical Reasoning (GSM-8K and MATH). #Params denote the number of trainable parameters. **bold** and underline represent the best and second best performing PEFT methods, respectively. SVFT offers superior/competitive performance at much lower #Params. For SVFT$_d^R$, we set $d = 16$ for Gemma and $d = 12$ for LLaMA-3 models.

| Method | Gemma-2B | | | Gemma-7B | | | LLaMA-3-8B | | |
|---|---|---|---|---|---|---|---|---|---|
| | #Params | GSM-8K | MATH | #Params | GSM-8K | MATH | #Params | GSM-8K | MATH |
| Full-FT | 2.5B | 52.69 | 17.94 | 8.5B | 74.67 | 25.70 | 8.0B | 64.13 | 16.24 |
| LoRA$_{r=32}$ | 26.2M | 43.06 | 15.50 | 68.8M | 76.57 | 29.34 | 56.6M | **75.89** | **24.74** |
| DoRA$_{r=16}$ | 13.5M | 44.27 | **16.18** | 35.5M | 74.52 | 29.84 | 29.1M | 75.66 | **24.72** |
| BOFT$_{m=2}^{b=8}$ | 1.22M | 36.01 | 12.13 | 2.90M | 71.79 | 28.98 | 4.35M | 67.09 | 21.64 |
| DoRA$_{r=1}$ | 1.19M | 35.25 | 13.04 | 3.26M | 74.37 | 26.28 | 2.55M | 68.30 | 21.96 |
| LoRA$_{r=1}$ | 0.82M | 32.97 | 13.04 | 0.82M | 72.4 | 26.28 | 1.77M | 68.84 | 20.94 |
| VeRA$_{r=1024}$ | 0.63M | 36.77 | 14.12 | 0.43M | 71.11 | 27.04 | 0.98M | 63.76 | 20.28 |
| SVFT$^P$ | 0.19M | 40.34 | 14.38 | 0.43M | 73.50 | 27.30 | 0.48M | 69.22 | 20.44 |
| SVFT$_d^R$ | 6.35M | **50.03** | 15.56 | 19.8M | **76.81** | **29.98** | 13.1M | **75.90** | 24.22 |

---

[4]BOFT is approximately three times slower than LoRA. The shared matrices in VERA can become a limiting factor for models with non-uniform internal dimensions, such as LLaMA-3.

Table 2: Evaluation results on eight commonsense reasoning benchmarks with Gemma-7B. We follow [19] for hyperparameter configurations, and report accuracy for all tasks. HS and WG denote HellaSwag [36] and WinoGrande [29], respectively. $\text{SVFT}^P$ offers competitive performance at a fraction of #Params. $\text{SVFT}_{d=8}^B$ can match $\text{LoRA}_{r=32}$ with $\sim$7x fewer parameters.

| Method | #Params | BoolQ | PIQA | SIQA | HS | WG | ARC-e | ARC-c | OBQA | Average |
|---|---|---|---|---|---|---|---|---|---|---|
| Full-FT | 8.5B | 72.32 | 87.32 | 76.86 | 91.07 | 81.76 | 92.46 | 82.76 | 89.00 | 84.19 |
| $\text{LoRA}_{r=32}$ | 68.8M | 71.55 | **87.95** | **77.27** | 91.80 | **79.71** | 92.67 | 82.16 | **86.40** | **83.69** |
| $\text{DoRA}_{r=16}$ | 35.5M | 71.46 | 87.59 | 76.35 | **92.11** | 78.29 | 92.00 | 80.63 | 85.60 | 83.00 |
| $\text{DoRA}_{r=1}$ | 3.31M | 68.22 | 86.72 | 75.23 | 91.14 | 78.13 | 91.87 | **83.19** | 86.20 | 82.59 |
| $\text{VeRA}_{r=2048}$ | 1.49M | 64.25 | 86.28 | 74.04 | 86.96 | 69.00 | 92.76 | 82.33 | 82.00 | 79.70 |
| $\text{LoRA}_{r=1}$ | 0.82M | 65.44 | 86.28 | 75.02 | 89.91 | 75.92 | 91.79 | 81.91 | 85.40 | 81.46 |
| $\text{SVFT}^P$ | 0.51M | 67.92 | 86.45 | 75.47 | 86.92 | 74.03 | 91.80 | 81.23 | 83.00 | 80.85 |
| $\text{SVFT}_{d=8}^B$ | 9.80M | **71.90** | 86.98 | 76.28 | 91.55 | 78.76 | **92.80** | 83.11 | 85.40 | 83.35 |

Table 3: DeBERTaV3$_\text{base}$ with different adaptation methods on the GLUE benchmark. We report matched accuracy for MNLI, Matthew's correlation for CoLA, Pearson correlation for STS-B, and accuracy for other tasks. Higher is better for all tasks. * indicates values reported in [20].

| Method | #Params | MNLI | SST-2 | MRPC | CoLA | QNLI | QQP | RTE | STS-B | Avg. |
|---|---|---|---|---|---|---|---|---|---|---|
| Full-FT* | 184M | 89.90 | 95.63 | 89.46 | 69.19 | 94.03 | **92.40** | 83.75 | 91.60 | 88.25 |
| $\text{LoRA*}_{r=8}$ | 1.33M | **90.65** | 94.95 | 89.95 | 69.82 | 93.87 | 91.99 | 85.20 | 91.60 | 88.50 |
| $\text{DoRA}_{r=4}$ | 0.75M | 89.92 | 95.41 | 89.10 | 69.37 | 94.14 | 91.53 | 87.00 | 91.80 | 88.53 |
| $\text{BOFT*}_{m=2}^{b=8}$ | 0.75M | 90.25 | **96.44** | **92.40** | **72.95** | 94.23 | 92.10 | **88.81** | **91.92** | **89.89** |
| $\text{LoRA}_{r=1}$ | 0.17M | 90.12 | 95.64 | 86.43 | 69.13 | 94.18 | 91.43 | 87.36 | 91.52 | 88.23 |
| $\text{VeRA}_{r=1024}$ | 0.09M | 89.93 | 95.53 | 87.94 | 69.06 | 93.24 | 90.4 | 87.00 | 88.71 | 87.73 |
| $\text{SVFT}^P$ | 0.06M | 89.69 | 95.41 | 88.77 | 70.95 | **94.27** | 90.16 | 87.24 | 91.80 | 88.54 |
| $\text{SVFT}_{d=2}^R$ | 0.28M | 89.97 | 95.99 | 88.99 | 72.61 | 93.90 | 91.50 | 88.09 | 91.73 | 89.10 |

# 5 Results

## 5.1 Performance on Language Tasks

**Natural Language Generation.** We present results on mathematical question answering against baseline PEFT techniques across three base models – varying from 2B to 8B parameters in Table 1. To ensure a comprehensive comparison, we test baseline techniques (LoRA, DoRA) with different configurations, and varying hyper-parameters like rank to cover a range of learnable parameters from low to high. Note that even when the rank is as low as 1, both methods yield more trainable parameters than $\text{SVFT}^P$. $\text{SVFT}^P$ ($\sim$0.2M) shows as much as $18\%$ relative improvement over techniques that use $6\times$ more trainable parameters ($\text{BOFT}_{m=2}^{b=8}$, $\text{LoRA}_{r=1}$). Against techniques of comparable size (VeRA), $\text{SVFT}^P$ achieves **15.5%** relative improvement on average. Even in the default regime, $\text{SVFT}_d^R$ matches techniques with at least $3\times$ more trainable parameters. Notably, on GSM-8K, $\text{SVFT}_d^R$ again achieves **96%** of full fine-tuning performance, while $\text{DoRA}_{r=16}$ recovers 86% with $2\times$ more parameters than $\text{SVFT}_d^R$.

**Commonsense Reasoning.** In Table 2, we compare performance on commonsense reasoning benchmarks with Gemma-7B, and observe similar trends. In the lower and moderately parameterized regime ($\sim$0.43M), $\text{SVFT}^P$ shows competitive performance in comparison to $\text{LoRA}_{r=1}$ and

Table 4: Performance on image classification benchmarks – CIFAR-100 (C100), Food101 (F101), Flowers102 (F102), and Resisc-45 (R45). We only adapt $Q, V$ matrices for all methods, following prior work [17]. We report accuracy for all tasks.

| Method | ViT Base | | | | | ViT Large | | | | |
|---|---|---|---|---|---|---|---|---|---|---|
| | #Params | C100 | F101 | F102 | R45 | #Params | C100 | F101 | F102 | R45 |
| Head | - | 78.58 | 75.14 | 98.71 | 64.42 | - | 79.14 | 75.66 | 98.89 | 64.99 |
| Full-FT | 85.8M | 85.02 | 75.41 | 99.16 | **75.30** | 303.3M | 87.37 | 78.67 | 98.88 | **80.17** |
| LoRA$_{r=8}$ | 294.9K | **85.65** | 76.13 | 99.14 | 74.01 | 786.4K | 87.36 | **78.95** | 99.24 | 79.55 |
| VeRA$_{r=256}$ | 24.6K | 84.00 | 74.02 | 99.10 | 71.86 | 61.4K | **87.55** | 77.87 | **99.27** | 75.92 |
| SVFT$^P$ | 18.5K | 83.78 | 74.43 | 98.99 | 70.55 | 49.2K | 86.67 | 77.47 | 99.09 | 73.52 |
| SVFT$^R_{d=4}$ | 165.4K | 84.85 | 76.45 | 99.17 | 74.53 | 441.5K | 87.05 | **78.95** | 99.23 | 78.90 |
| SVFT$^B_{d=4}$ | 165.4K | 84.65 | **76.51** | **99.21** | 75.12 | 441.5K | 86.95 | 78.85 | 99.24 | 78.93 |

DoRA$_{r=1}$, which have $1.9\times$ and $7.7\times$ more parameters, respectively. Against VeRA, which trains $3.5\times$ more parameters, SVFT$^P$ shows a relative improvement of $\sim$**1.16%**. Similarly, SVFT$^B_{d=8}$ also matches or exceeds methods that use up to $7\times$ more trainable parameters. For instance, SVFT$^B_{d=8}$ attains an average performance of 83.35% with only 9.8M parameters, closely matching LoRA$_{r=16}$ (83.69%, 68.8M parameters). We observe similar trends with Gemma-2B (refer Table 11).

**Natural Language Understanding.** Results on the GLUE benchmark are summarized in Table 3. SVFT matches LoRA$_{r=8}$ and DoRA$_{r=4}$ which use **12-22$\times$** more trainable parameters. Similarly, when compared to OFT and BOFT, SVFT$^P$ maintains a comparable average performance despite being $12\times$ smaller. These results highlight SVFT's ability to strike a balance between parameter efficiency and performance, making it an attractive PEFT choice for simple classification tasks.

**Parameter efficiency.** In Figure 1, we plot the performance of SVFT on mathematical reasoning and commonsense reasoning against other PEFT techniques across a range of configurations. Across trainable parameter budgets ranging from lowest to highest, SVFT obtains the best overall performance, matching methods that require significantly more trainable parameters. These results establish SVFT as a pareto-dominant approach for parameter-efficient fine-tuning.

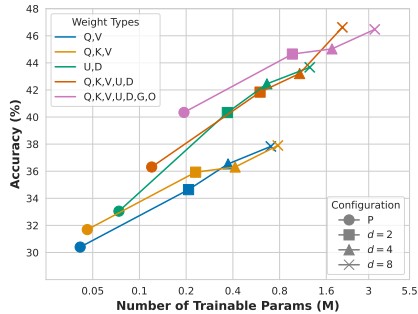

Figure 4: Performance variation with SVFT$^B_d$ based on the adapted weight matrices – GSM-8K with Gemma-2B. Adapting more target weight types results in greater gains in performance. Interestingly, for a fixed parameter budget, adapting $U$ and $D$ weight types gives greater lifts than adapting $Q$ and $V$.

## 5.2 Performance on Vision Tasks

Table 4 presents a comparison between SVFT and other PEFT techniques on image classification benchmarks, using ViT-B and ViT-L models. The results show that SVFT variants achieve a strong balance between performance and parameter efficiency, often surpassing or matching the performance of other methods with fewer learnable parameters. Notably, the SVFT$^B$ variant attains an average accuracy of $83.87\%$ across tasks with ViT-Base, outperforming Full-FT, which achieves a close $83.72\%$. Additionally, it's important to note that in these vision experiments, both classifier head parameters and other learnable parameters are trained.

## 5.3 Memory Analysis

Although SVFT reduces trainable parameters, it results in higher overall GPU memory usage compared to LoRA. However, fewer trainable parameters lower the memory demands for gradients,

activities, optimizer states, and other buffers. To validate this, we used HuggingFace's internal memory profiler to measure peak GPU memory usage. Our results, along with the adapted modules for all baselines, are summarized in Table 5. We observe that SVFT uses approximately 1.2x more memory than LoRA but remains comparable to or more efficient than DoRA. We present additional analysis in Appendix C.5.

Table 5: GPU Memory analysis, measured in gigabytes (GB). We report the average performance on GSM-8K and MATH. SVFT outperforms both LoRA and DoRA in terms of performance while requiring lesser GPU memory than DoRA.

| Method | Target Modules | Gemma-2B | | | Gemma-7B | | |
|---|---|---|---|---|---|---|---|
| | | #Params | GPU Mem | Perf. | #Params | GPU Mem | Perf. |
| $\text{LoRA}_{r=4}$ | Q,K,V,U,D | 3.28M | 18.88 | 27.56 | 8.6M | 63.57 | 51.03 |
| $\text{DoRA}_{r=4}$ | Q,K,V,U,D | 3.66M | 24.58 | 28.44 | 9.72M | 78.70 | 51.94 |
| $\text{LoRA}_{r=32}$ | Q,K,V,U,D | 26.2M | 19.06 | 29.28 | 68.8M | 64.24 | 52.96 |
| $\text{DoRA}_{r=16}$ | Q,K,V,U,D | 13.5M | 24.64 | 30.22 | 35.5M | 78.99 | 52.18 |
| $\text{SVFT}^P$ | Q,K,V,U,D,O,G | 194K | 21.90 | 27.36 | 429K | 76.26 | 50.40 |
| $\text{SVFT}^R_{d=8}$ | Q,K,V,U,D,O,G | 3.28M | 22.02 | 31.87 | 9.8M | 76.65 | 50.99 |
| $\text{SVFT}^R_{d=16}$ | Q,K,V,U,D,O,G | 6.35M | 22.15 | **32.79** | 19.8M | 77.04 | **53.40** |

## 5.4 Contribution of Each Weight Type

In Figure 4, we investigate the contribution of each weight type. Starting with the base configuration, we apply $\text{SVFT}^B_d$ to the $Q$ and $V$ weights in each transformer block and report the performance. We then incrementally add the remaining weight modules ($K, U, D, O, G$) and observe the changes in performance. For each configuration, we also vary the trainable parameters by incrementing the total learnable off-diagonals.

Note that applying $\text{SVFT}^B_d$ to $U, D, O$, and $G$ does not increase trainable parameters as much as applying LoRA/DoRA to these modules (Table 8). For example, for a large matrix of shape $d_1 \times d_2$, $\text{LoRA}_{r=1}$ learns $d_1 + d_2$ parameters, while $\text{SVFT}^P$ learns $\min(d_1, d_2)$ parameters. We observe that adapting only $U$ and $D$ with SVFT yields up to a $10\%$ relative improvement over adapting $Q$ and $V$ for the same parameter budget ($\sim 0.8M$). Our findings indicate that adapting more weight types enhances performance.

## 5.5 Impact of $M$'s Structure on Performance

We analyze the impact of various parameterizations of $M$ (Plain, Banded, Random, Top-$k$) on downstream performance. To ensure a fair comparison, we align the number of trainable coefficients across all variants whenever possible. As shown in Table 7, the Banded variant outperforms the others, followed closely by the Random variant, across different models and tasks. This trend is also evident in the average rank column of the table. Based on these empirical findings, we recommend using the Banded variant.

## 5.6 Impact of Pre-trained Weight Quality

A key feature of SVFT is that the weight update depends on the pre-trained weights $W$. We therefore ask the following question: *Does the quality of pre-trained weights have a disproportionate impact on* SVFT*?* To

Table 6: Results on GSM-8K after fine-tuning on Pythia-2.8B checkpoints at different stages of pre-training (PT).

| Method | #Params | PT Steps | | $\Delta$Perf |
|---|---|---|---|---|
| | | 39K | 143K | |
| Full-FT | 2.5B | 21.00 | 30.09 | 9.09 |
| LoRA | 5.24M | 11.22 | 18.95 | 7.73 |
| SVFT | 5.56M | 15.08 | 23.19 | 8.11 |

Table 7: Results on fine-tuning with SVFT using different $M$ parameterizations.

| Structure | Gemma-2B | | | Gemma-7B | | | LLaMA-3-8B | | | Avg. Rank |
|---|---|---|---|---|---|---|---|---|---|---|
| | #Params | GSM-8K | MATH | #Params | GSM-8K | MATH | #Params | GSM-8K | MATH | |
| Plain | 0.2M | 40.34 | 14.38 | 0.43M | 73.50 | 27.30 | 0.48M | 69.22 | 20.44 | 4 |
| Banded | 6.4M | 47.84 | **15.68** | 19.8M | **76.81** | **29.98** | 17.2M | **75.43** | **24.44** | **1** |
| Random | 6.4M | **50.03** | 15.56 | 19.8M | 76.35 | 29.86 | 17.2M | 74.07 | 23.78 | 2 |
| Top-$k$ | 6.4M | 49.65 | 15.32 | 19.8M | 76.34 | 29.72 | 17.2M | 73.69 | 23.96 | 3 |

answer this, we consider two checkpoints from the Pythia suite [2] at different stages of training, i.e., 39K steps and 143K steps, respectively. We fine-tune each of these checkpoints independently with Full-FT, LoRA, and SVFT. We then compare the increase in performance ($\Delta$Perf). As shown in Table 6, compared to LoRA, SVFT benefits more from better pre-trained weights. We also note that SVFT outperforms LoRA in both settings, suggesting that the benefits of inducing a $\Delta W$ that explicitly depends on $W$ are beneficial even when $W$ is sub-optimal.

## 6 Discussion

**Limitations.** Despite significantly reducing learnable parameters and boosting performance, SVFT incurs some additional GPU memory usage. Unlike LoRA, SVFT necessitates computing the SVD and storing both left and right singular vectors. While memory consumption remains comparable to or lower than DoRA and BOFT, it's roughly $1.2\times$ that of LoRA. However, similar to the scaling explored in [34], memory usage should amortize with the increasing scale of adaptation tasks. In future work we will explore quantization and other techniques to address memory concerns.

**Broader Impact.** Our work enables easier personalization of foundational models, which can have both positive and negative societal impacts. Since our method provides computational efficiency (smaller parameter footprint), it will be less expensive to enable personalization.

## 7 Conclusion

This work introduces SVFT, a novel and efficient PEFT approach that leverages the structure of pre-trained weights to determine weight update perturbations. We explore four simple yet effective sparse parameterization patterns, offering flexibility in controlling the model's expressivity and the number of learnable parameters. Extensive experiments on language and vision tasks demonstrate SVFT's effectiveness as a PEFT method across diverse parameter budgets. Furthermore, we theoretically show that SVFT can induce higher-rank perturbation updates compared to existing methods, for a fixed parameter budget. In future work, we aim to develop principled methods to generate sparsity patterns, potentially leading to further performance improvements.

## Acknowledgements

We would like to thank Greg Kuhlmann for helping support this research. This work was also supported by the NSF institutes ENCORE and IFML.

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

## Appendix

The appendix is organized as follows.

- In Appendix A, we give proofs for the lemmas outlined in 3.2.
- In Appendix B, we compare how the trainable parameters count for different PEFT techniques (LoRA, DoRA, VeRA) versus our method SVFT.
- In Appendix C, we describe results for additional experiments and provide implementation details for all the experiments.

## A    Proofs

We provide brief proofs for the *Structure*, *Expressivity* and the *Rank* lemmas for SVFT:

*(a) Structure:* If $M$ is diagonal, then the final matrix $W_0 + UMV^T$ can be written as $U(\Sigma + M)V^T$ since $W_0 = U\Sigma V^T$, where $(\Sigma + M)$ is also a diagonal matrix. Thus, $U(\Sigma + M)V^T$ is a valid and unique SVD of $W_0 + UMV^T$ up to sign flips in the singular vectors.

*(b) Expressivity:* Finding $M$ for any target matrix $P$ of size $d_1 \times d_2$ such that $P = W_0 + UMV^T$ is the same as finding $M$ for a new target matrix $P' = P - W_0$ such that $P' = UMV^T$. For a full SVD, the dimension of $M$ is $d_1 \times d_2$ and since the dimension of $P'$ is also $d_1 \times d_2$, $P' = UMV^T$ is a bijection and $M = U^T(P - W_0)V$ (since $U$ and $V$ are orthogonal).

*(c) Rank:* If $M$ has $k$ non-zero elements, then the rank of the update $UMV^T$ will be upper bounded by $k$ (since by Gaussian elimination, $k$ or less elements will remain, the best case being all $k$ elements in the diagonal). We also know that the rank is upper bounded by $\min\{d_1, d_2\}$, giving an achievable upper bound on the rank as $\min\{k, \min\{d_1, d_2\}\}$.

## B    Parameter Count Analysis

Table 8: Parameter count analysis. $L_{\text{tuned}}$, $D_{\text{model}}$, $r$, $k$ denote total layers being adapted, hidden dimension, rank, and additional off-diagonals respectively.

| Method | Trainable Parameter Count |
|--------|---------------------------|
| LoRA | $2 \times L_{\text{tuned}} \times D_{\text{model}} \times r$ |
| DoRA | $L_{\text{tuned}} \times D_{\text{model}} \times (2r + 1)$ |
| VeRA | $L_{\text{tuned}} \times (D_{\text{model}} + r)$ |
| SVFT$^P$ | $L_{\text{tuned}} \times D_{\text{model}}$ |
| SVFT$^B_{d=k}$ | $L_{\text{tuned}} \times (D_{\text{model}} \times k + (D_{\text{model}} - k)(k + 1))$ |

## C    Additional Experiments and Implementation Details

All of our experiments are conducted on a Linux machine (Debian GNU) with the following specifications: $2 \times$A100 80 GB, Intel Xeon CPU @ 2.20GHz with 12 cores, and 192 GB RAM. For all our experiments (including baseline experiments), we utilize hardware-level optimizations like mixed weight precision (e.g., bfloat16) whenever possible.

### C.1    Comparison against SVD-based Variants

We compare SVF [31] and our proposed method, SVFT, on the GSM-8K and MATH benchmarks using Gemma-2B. The results are presented in Table 9. The results indicate that SVF and SVFT$^P$ exhibit comparable performance on these benchmarks, as expected due to their design equivalence. This

finding also applies to SVDiff [10] and SAM-parser [25] for the same reason. Additionally, the table highlights a significant performance improvement when comparing SVF to SVFT$^R$, demonstrating the advantage of learning the off-diagonal elements.

Table 9: Results of SVF and SVFT on GSM-8K and MATH with Gemma-2B.

| Method | #Params | Target Modules | GSM-8K | MATH |
|---|---|---|---|---|
| SVF | 120K | Q,K,V,U,D | 36.39 | 13.96 |
| SVFT$^P$ | 120K | Q,K,V,U,D | 36.39 | 13.86 |
| SVF | 194K | Q,K,V,U,D,O,G | 39.12 | 14.02 |
| SVFT$^P$ | 194K | Q,K,V,U,D,O,G | 40.34 | 14.38 |
| SVFT$^R_{d=16}$ | 6.35M | Q,K,V,U,D,O,G | 50.03 | 15.56 |

## C.2 Performance Evaluation with Fixed Target Module Adaptation

We compare SVFT to baseline methods, adapting the same target modules to ensure a consistent evaluation. Results are presented in Table 10, showing that SVFT outperforms other methods in this setup.

Table 10: Performance (Accuracy) on Mathematical Reasoning (GSM-8K and MATH). All methods are applied on the target modules {Q,K,V,U,D} with SVFT demonstrating superior performance. When applying SVFT$^R$ on Gemma-2B and LLaMA-3-8B we use $d = 12$ and $d = 24$ respectively.

| Method | Gemma-2B | | | LLaMA-3-8B | | |
|---|---|---|---|---|---|---|
| | #Params | GSM-8K | MATH | #Params | GSM-8K | MATH |
| LoRA$_{r=4}$ | 3.28M | 40.60 | 14.50 | 7.07M | 69.37 | 22.90 |
| DoRA$_{r=4}$ | 3.66M | 41.84 | 15.04 | 7.86M | 74.37 | 24.10 |
| LoRA$_{r=32}$ | 26.2M | 43.06 | 15.50 | 56.6M | **75.89** | 24.74 |
| DoRA$_{r=16}$ | 13.5M | 44.27 | 16.18 | 29.1M | 75.66 | 24.72 |
| SVFT$^R$ | 2.98M | **47.41** | **16.72** | 15.98M | 75.32 | **25.08** |

## C.3 Commonsense Reasoning Gemma-2B

We evaluate and compare SVFT variants against baseline PEFT methods on commonsense reasoning tasks with Gemma-2B model and tabulate results in Table 11.

## C.4 Are All Singular Vectors Important?

To determine the importance of considering all singular vectors and singular values during fine-tuning, we reduce the rank of $U$ and $V$, and truncate $\Sigma$ and $M$ to an effective rank of $r$. If the original weight matrix $W \in \mathbb{R}^{m \times n}$, then after truncation, $U \in \mathbb{R}^{m \times r}, V \in \mathbb{R}^{n \times r}$. This truncation significantly reduces the number of trainable parameters, so we compensate by increasing the number of off-diagonal coefficients ($d$) in $M$.

Table 11: Results with Gemma-2B on eight commonsense reasoning benchmarks. We follow [19] for hyperparameter configurations, and report accuracy for all tasks.

| Method | #Params | BOOLQ | PIQA | SIQA | HellaSwag | Winogrande | ARC-E | ARC-C | OBQA | Average |
|---|---|---|---|---|---|---|---|---|---|---|
| Full-FT | 2.5B | 63.57 | 74.1 | 65.86 | 70.00 | 61.95 | 75.36 | 59.72 | 69 | 67.45 |
| LoRA$_{r=32}$ | 26.2M | 63.11 | 73.44 | 63.20 | 47.79 | 52.95 | 74.78 | 57.16 | 67.00 | 62.43 |
| LoRA$_{r=16}$ | 13.5M | 62.87 | 73.93 | 65.34 | 53.16 | 55.51 | 76.43 | 59.55 | 68.4 | 64.40 |
| BOFT$_{m=2}^{b=8}$ | 1.22M | 59.23 | 63.65 | 47.90 | 29.93 | 50.35 | 59.04 | 42.66 | 41.00 | 49.22 |
| VeRA$_{r=2048}$ | 0.66M | 62.11 | 64.31 | 49.18 | 32.00 | 50.74 | 58.08 | 42.83 | 42.6 | 50.23 |
| LoRA$_{r=1}$ | 0.82M | 62.2 | 69.31 | 56.24 | 32.47 | 51.53 | 69.52 | 48.8 | 56.4 | 55.81 |
| DoRA$_{r=1}$ | 1.19M | 62.17 | 68.77 | 55.93 | 32.95 | 51.22 | 68.81 | 48.72 | 55.6 | 55.52 |
| SVFT$^P$ | 0.19M | 62.26 | 70.18 | 56.7 | 32.47 | 47.04 | 69.31 | 50.08 | 58.4 | 55.81 |
| SVFT$_{d=16}^B$ | 6.35M | 63.42 | 73.72 | 63.86 | 71.21 | 59.58 | 73.69 | 54.77 | 66.6 | 65.86 |

Table 12: Performance with varying rank ($r$) and the off-diagonal elements ($d$) of $M$. When $r = 2048$, the update is full-rank.

| Rank ($r$) | Diags ($d$) | #Params | GSM-8K | MATH |
|---|---|---|---|---|
| 128 | 64 | 1.55M | 0.98 | 0.21 |
| 1536 | - | 0.15M | 16.37 | 3.64 |
| 1536 | 2 | 0.74M | 25.01 | 6.04 |
| 2048 | - | 0.19M | **40.34** | **14.38** |

Our results, with four different configurations of $r$ and $d$, are presented in Table 12. The findings show that a very low rank ($r = 128$) leads to poor performance, even when parameters are matched. A reasonably high rank of $r = 1536$, which is 75% of the full rank, still fails to match the performance of the full-rank variant that has $0.25\times$ the trainable parameters. This indicates that all singular vectors significantly contribute to the end task performance when fine-tuning with SVFT, and that important information is lost even when truncating sparingly.

## C.5 Additional Memory Analysis Experiments

We present additional memory analysis experiments for Gemma-2B and LLaMA-3-8B in Table 13 and Table 14. SVFT variants consume lesser memory than DoRA and $1.2\times$ more memory than LoRA.

Table 13: Memory analysis for Gemma-2B.

| Method | Target Modules | #Params | GPU Mem (GB) | GSM-8K | MATH |
|---|---|---|---|---|---|
| LoRA$_{r=4}$ | Q,K,V,U,D | 3.28M | 18.88 | 40.63 | 14.5 |
| DoRA$_{r=4}$ | Q,K,V,U,D | 3.66M | 24.58 | 41.84 | 15.04 |
| LoRA$_{r=32}$ | Q,K,V,U,D | 26.2M | 19.06 | 43.06 | 15.5 |
| DoRA$_{r=16}$ | Q,K,V,U,D | 13.5M | 24.64 | 44.27 | 16.18 |
| SVFT$_{d=12}^R$ | Q,K,V,U,D | 2.98M | 20.50 | 47.61 | **16.72** |
| SVFT$^P$ | Q,K,V,U,D,O,G | 194K | 21.90 | 40.34 | 14.38 |
| SVFT$_{d=8}^R$ | Q,K,V,U,D,O,G | 3.28M | 22.02 | 47.76 | 15.98 |
| SVFT$_{d=16}^R$ | Q,K,V,U,D,O,G | 6.35M | 22.15 | **50.03** | 15.56 |

Table 14: Memory analysis for LLaMA-3-8B.

| Method | Target Modules | #Params | GPU Mem (GB) | GSM-8K | MATH |
|---|---|---|---|---|---|
| $\text{LoRA}_{r=1}$ | Q,K,V,U,D | 1.77M | 57.82 | 68.84 | 20.94 |
| $\text{DoRA}_{r=1}$ | Q,K,V,U,D | 2.56M | 70.17 | 68.30 | 21.96 |
| $\text{LoRA}_{r=32}$ | Q,K,V,U,D | 56.6M | 58.41 | 75.89 | 24.74 |
| $\text{DoRA}_{r=16}$ | Q,K,V,U,D | 29.1M | 70.44 | 75.66 | 24.72 |
| $\text{SVFT}^B_{d=24}$ | Q,K,V,U,D | 15.98M | 71.52 | 75.32 | **25.08** |
| $\text{SVFT}^R_{d=12}$ | U,D,O,G | 13.1M | 70.37 | **75.90** | 24.22 |
| $\text{SVFT}^P$ | Q,K,V,U,D,O,G | 483K | 73.95 | 69.22 | 20.44 |
| $\text{SVFT}^R_{d=12}$ | Q,K,V,U,D,O,G | 15.9M | 71.52 | 73.99 | **25.08** |

## C.6 Performance vs Total Trainable Parameters

In addition to the experiments performed in Figure 1 (main paper) for Gemma-2B on challenging natural language generation (NLG) tasks like GSM-8K and Commonsense Reasoning, we also plot the performance vs total trainable parameters for larger state-of-the-art models like Gemma-7B and LLaMA-3-8B on GSM-8K. Figure 5 further demonstrates SVFT's pareto-dominance. On larger models, we observe that full-finetuning overfits, leading to sub-optimal performance in comparison to PEFT methods.

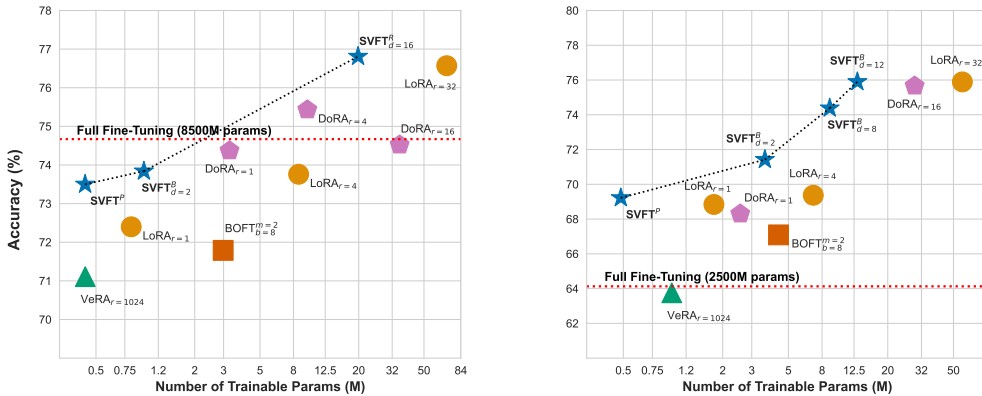

Figure 5: Performance versus total trainable parameters for GSM-8K on Gemma-7B (left) and LLaMA-3-8B (right).

## C.7 Settings for Language Tasks

**Natural Language Understanding.** We fine-tune the DeBERTaV3$_{\text{base}}$ [11] model and apply SVFT to all linear layers in every transformer block of the model. We only moderately tune the batch size, learning rate, and number of training epochs. We use the same model sequence lengths used by [20] to keep our comparisons fair. The hyperparameters used in our experiments can be found in Table 15.

**Natural Language Generation.** See the hyperparameters used in our experiments in Table 16. For LoRA, DoRA, we adapt $Q, K, V, U, D$ matrices. We apply BOFT on $Q, V$ matrices since applying on multiple modules is computationally expensive. For VeRA, which enforces a constraint of uniform internal dimensions for shared matrices, we apply on $G, U$ projection matrices as it yields the highest number of learnable parameters. We apply SVFT on $Q, K, V, U, D, O, G$ for the Gemma family of models, and $U, D, O, G$ for LLaMA-3-8B. Note that applying SVFT on these modules does not increase trainable parameters at the same rate as applying LoRA or DoRA on them would. We adopt the code base from https://github.com/meta-math/MetaMath.git for training scripts and

Table 15: Hyperparameter setup used for DeBERTaV3$_{base}$ on the GLUE benchmark.

| Method | Dataset | MNLI | SST-2 | MRPC | CoLA | QNLI | QQP | RTE | STS-B |
|---|---|---|---|---|---|---|---|---|---|
| | Optimizer | | | | AdamW | | | | |
| | Warmup Ratio | | | | 0.1 | | | | |
| | LR Schedule | | | | Linear | | | | |
| | Learning Rate (Head) | | | | 6E-03 | | | | |
| | Max Seq. Len. | 256 | 128 | 320 | 64 | 512 | 320 | 320 | 128 |
| | # Epochs | 10 | 10 | 30 | 20 | 10 | 6 | 15 | 15 |
| SVFT$^P$ | Batch Size | 32 | 32 | 16 | 16 | 32 | 16 | 4 | 32 |
| | Learning Rate | 5E-02 | 5E-02 | 5E-02 | 8E-02 | 8E-02 | 5E-02 | 5E-02 | 5E-02 |
| SVFT$^R_{d=2}$ | Batch Size | 32 | 32 | 16 | 16 | 32 | 32 | 16 | 32 |
| | Learning Rate | 1E-02 | 1E-02 | 1E-02 | 1E-02 | 3E-02 | 1E-02 | 3E-02 | 1E-02 |

evaluation setups and use the fine-tuning data available at `https://huggingface.co/datasets/meta-math/MetaMathQA-40K`.

Table 16: Hyperparameter setup used for fine-tuning on MetaMathQA-40K.

| Hyperparameter | Gemma-2B | | Gemma-7B | | LLaMA-3-8B | |
|---|---|---|---|---|---|---|
| | SVFT$^P$ | SVFT$^R_{d=16}$ | SVFT$^P$ | SVFT$^R_{d=16}$ | SVFT$^P$ | SVFT$^R_{d=12}$ |
| Optimizer | | | AdamW | | | |
| Warmup Ratio | | | 0.1 | | | |
| LR Schedule | | | Cosine | | | |
| Learning Rate | 5E-02 | 1E-03 | 5E-02 | 1E-03 | 5E-02 | 1E-03 |
| Max Seq. Len. | | | 512 | | | |
| # Epochs | | | 2 | | | |
| Batch Size | | | 64 | | | |

**Commonsense Reasoning.** See the hyperparameters used in our experiments in Table 17. We adopt the same set of matrices as that of natural language generation tasks. We use the code base from `https://github.com/AGI-Edgerunners/LLM-Adapters`, which also contains the training and evaluation data.

## C.8 Settings for Vision Tasks

For each dataset in the vision tasks, we train on 10 samples per class, using 2 examples per class for validation, and test on the full test set. Similar to previous literature, we always train the classifier head for these methods since the number of classes is large. The parameter counts do not include the number of parameters in the classification head. The hyperparameters are mentioned in Table 18. We tune the learning rates for SVFT and BOFT select learning rates for other methods from [17], run training for 10 epochs, and report test accuracy for the best validation model. For all methods, since the classification head has to be fully trained, we report the parameter count other than the classification head.

Table 17: Hyperparameter setup used for fine-tuning on commonsense-15K.

| Hyperparameter | Gemma-2B | | Gemma-7B | |
|---|---|---|---|---|
| | $\text{SVFT}^P$ | $\text{SVFT}^B_{d=8}$ | $\text{SVFT}^P$ | $\text{SVFT}^B_{d=8}$ |
| Optimizer | AdamW | | | |
| Warmup Steps | 100 | | | |
| LR Schedule | Linear | | | |
| Max Seq. Len. | 512 | | | |
| # Epochs | 3 | | | |
| Batch Size | 64 | | | |
| Learning Rate | 5E-02 | 5E-03 | 5E-02 | 1E-03 |

Table 18: Hyperparameter setup used for fine-tuning on all vision tasks.

| Hyperparameter | ViT-B    ViT-L |
|---|---|
| Optimizer | AdamW |
| Warmup Ratio | 0.1 |
| Weight Decay | 0.01 |
| LR Schedule | Linear |
| # Epochs | 10 |
| Batch Size | 64 |
| $\text{SVFT}^P$ Learning Rate (Head) | 4E-03 |
| $\text{SVFT}^P$ Learning Rate | 5E-02 |
| $\text{SVFT}^B_{d=2}$ Learning Rate (Head) | 4E-03 |
| $\text{SVFT}^B_{d=2}$ Learning Rate | 5E-02 |
| $\text{SVFT}^B_{d=8}$ Learning Rate (Head) | 4E-03 |
| $\text{SVFT}^B_{d=8}$ Learning Rate | 5E-03 |

