# OpenReview forum: "SVFT: Parameter-Efficient Fine-Tuning with Singular Vectors"
_NeurIPS.cc/2024/Conference — NeurIPS 2024 poster_

### Official Review · Reviewer_DEin · 2024-06-17

**Soundness:** 3
**Presentation:** 3
**Contribution:** 3
**Rating:** 6
**Confidence:** 4

**Summary:**

- Proposes method for parameter efficient fine-tuning using Singular Value Decomposition of the pretrained weights
- Fine-tuning is done by adding a learned residual $\Delta W = UMV^\top$, where $M$ is a sparse trainable matrix, which results in fine-tuned weigts $\hat{W} = U\Sigma V^\top + UMV^{\top}$
- Proposes 4 variants for sparsity: 1) Plain: $M$ is a diagonal matrix 2) Banded: the off-diagonal elements of $M$ are non-zero in a banded pattern 3) Random: Randomly selects $k$ elements of $M$ to be non-zero 4) Top-k: Selects tok $k$ elements for which the alingement between left and rght eigenvectors is the highest
- Provides theoretical insights on the properties of the method concerning the structure, expressivity and rank of the fine-tuning residuals

**Strengths:**

1. The idea of using SVD of the pretrained weights for parameter efficient fine-tuning is interesting.
2. The number of trainable parameters increases at a lower rate than LoRA/DoRA if more layers are adapted.
3. Provides theoretical insights on the properties of the method concerning the structure, expressivity and rank of the fine-tuning residuals
4. Shows better performance compared to VeRA, which suggests that SVFT is better for reducing the number of trainable parameters that the random parameters of VeRA (at the cost of higher memory usage). It also does not have a restriction of requiring same dimensions across layers like VeRA.
5. Observation that truncating the singular vectors leads to loss of performance in Appendix C.3 is interesing and informative.
5. The code is provided for verification.

**Weaknesses:**

1. **Missing citation and comparison with prior work**:
    1) The proposed method is very similar to [1], which has not been cited. [1] fine-tunes the singular values (rather than a residual addition) for few shot semantic segmentation tasks. Given how closely related SVFT $^P$ is to [1] (even though the original work was in context of segmentation methods), a comparison should be performed on the benchmarks in this paper. This work is different enough, and does show additional insights, but a comparison should be performed for completeness.
2. **Clarity and coherence of writing**:
    1) In Table 3 (NLU experiments on GLUE), the paper mentions that the results fo some baseline methods are obtained from prior works, but does not cite the actual works that the results are obtained from
    2) The paper mentions the components chosen for adaptation for SVFT (in the Appendix) but does not mention anything about the baseline methods. This makes it hard for a reader to know the details of the implementation up front, and also makes the comparison between the methods difficult.
    3) On lines 14, 15, 16 and lines 46, 47, 48, the exact benchmark and model that generate these results should be mentioned - otherwise it's difficult to follow the paper.
    4) The weight types adapted for each benchmark is different, and is not consistently mentioned in the main text, which makes it hard to follow the paper.
3. **Increased memory usage during training**
    1) While it's a positive that SVFT can keep the parameter count down while adapting more weight types, this will come at the cost of increased memory requirements.
    2) The advantage of reducing the parameters is a reduction in the memory usage due to the gradients (first and second order moments in case of AdamW). It might be possible that the actual memory usage is still comparable to LoRA if only a small number of layers are adapted (this can be verified by checking the actual memory allocated on the GPU during training) given that the modules chosen for adaptation are the same. However, if SVFT requires more weight types to be fine-tuned than LoRA/DoRA, the memory usage will be higher even if the number of parameters is lower. Hence, for all the methods, the actual memory usage must be reported for the chosen weight types the methods are applied to.
4. **Inconsistencies in evaluations**:
    1) Natural Language Generation: For LoRA and DoRA, Q,K,V,U,D matrices are adapted (which is inline with [3]). However, for SVFT, Q,K,V,U,D,O,G are adapted for the Gemma family of models, and U,D,O,G for LLaMA-3-8B. This means that the performance cannot be attributed to the method rather than the choice of weight types adapted.
    2) To decouple the choice of weight types adapted from the method, a comparison with LoRA and DoRA should be performed by adapting the same weight types. The performance of SVFT when Q,K,V,U,D are adapted for all the models must be compared with LoRA and DoRA.
    3) Relating to the point above, for SVFT, why are U,D,O,G adapted for for Llama 3-8B and Q,K,V,U,D,O,G for Gemma on the Math benchmark?
    4) With these inconsistent evaluation choices, the effectiveness, advantages and disagvantages of the method are not clear. If SVFT works better than baselines for certain components and not others, the paper should reflect this - and also show the memory required when using the weight types chosen for the respective methods.
5. **The results on Vision tasks are confusing**:
    1) The results with ViT-B and ViT-L in the main text are with different benchmarks (with ViT-B: CIFAR100 and Flowers102, and with ViT-L: Food101 and Resisc45).
    2) On closer inspection of Table 9 in the Appendix, SVFT variants perform worse than LoRA and DoRA on Resisc45 with ViT-B, and marginally better on Food101. This suggests that SVFT is not as effective as LoRA and DoRA on some benchmarks even though it reduces the parameters. This should be mentioned in the main text.
    3) The paper replicates the settings used in [2]. However, [2] adapts the Q and V layers for LoRA, which results in a high performance at a much lower parameter cost (comparison in the Table below, taken from this submission, and from [2]) than what the authors report with their implementation of LoRA applied to all linear layers. This can mean two things:
        1) The training data/checkpoints are different, which can cause a difference in performance
        2) Tuning all the linear layers is not a good idea for LoRA based on these results and [2], and is not indicative of LoRA's actual performance which makes the comparison unfair
    4) To have a proper comparison with baselines, the above two points need to be disambiguated by testing how LoRA performs by adapting Q and V with the training data used by the authors for this submission. The arguments made for NLG in point 4 are also applicable here.


| Method     | #Params | CIFAR100 | Flowers102 | Food101 | Resisc45 | #Params | CIFAR100 | Flowers102 | Food101 | Resisc45 |
|------------|---------|----------|------------|---------|----------|---------|----------|------------|---------|----------|
|            |         | ViT-B    |            |         |          |         | ViT-L    |            |         |          |
| LoRA $_{r=8}$ | 1.32M   | 84.41    | 99.23      | 76.02   | 76.86    | 0.35M   | 86.00    | 97.93      | 77.13   | 79.62    |
| DoRA $_{r=8}$ | 1.41M   | 85.03    | 99.30      | 75.88   | 76.95    | 3.76M   | 83.55    | 98.00      | 76.41   | 78.32    |
| LoRA $_{r=1}$ | 0.16M   | 84.86    | 96.88      | 73.35   | 76.33    | 0.44M   | 85.97    | 98.28      | 75.97   | 78.02    |
| DoRA $_{r=1}$ | 0.25M   | 84.46    | 99.15      | 74.80   | 77.06    | 0.66M   | 84.06    | 98.11      | 75.90   | 78.02    |
| VeRA       | 24.6K   | 83.38    | 98.59      | 75.99   | 70.43    | 61.4K   | 86.77    | 98.94      | 75.97   | 72.44    |
| SVFT $^{P}$    | 18.5K   | 83.85    | 98.93      | 75.68   | 67.19    | 49.2K   | 86.74    | 97.56      | 75.95   | 71.97    |
| SVFT $_{d=2}$ | 0.28M   | 84.72    | 99.28      | 75.64   | 72.49    | 0.74M   | 86.59    | 98.24      | 77.94   | 79.70    |
| SVFT $^{B=4}_{d=4}$ | 0.50M   | 83.17    | 98.52      | 76.54   | 66.65    | 1.32M   | 87.10    | 97.71      | 76.67   | 71.10    |
| SVFT $_{d=8}$ | 0.94M   | 85.69    | 98.88      | 76.70   | 70.41    | 2.50M   | 87.26    | 97.89      | 78.36   | 73.83    |
| From [2]           |         | ViT-B    |            |         |          |         | ViT-L    |            |         |          |
| LoRA $_{r=8}$ | 0.294M   | 85.9    | 98.8      | 89.9   | 77.7    | 0.786M   | 87.00    | 99.91      | 79.5   | 78.3    |
| VeRA       | 24.6K   | 84.8    | 99.0     | 89.0   | 77.0    | 61.4K   | 87.5    | 99.2      | 79.2   | 78.6    |


Following studies are needed to verify the effectiveness of the method:
1) A comparison with [1] (I would expect this to perform similar to SVFT $^P$, and any one benchmark, say Math, should be enough to verify this).
2) Reporting the memory usage - Point 3 in Weaknesses
2) Ablation study for point 4 (ii) in Weaknesses.
3) Verification for point 5 (iii) in Weaknesses.

**References**:

[1] Yanpeng Sun, Qiang Chen, Xiangyu He, Jian Wang, Haocheng Feng, Junyu Han, Errui Ding, Jian Cheng, Zechao Li, Jingdong Wang: Singular Value Fine-tuning: Few-shot Segmentation requires Few-parameters Fine-tuning. NeurIPS 2022

[2] Dawid Jan Kopiczko, Tijmen Blankevoort, Yuki Markus Asano: VeRA: Vector-based Random Matrix Adaptation. ICLR 2024

[3] Shih-Yang Liu, Chien-Yi Wang, Hongxu Yin, Pavlo Molchanov, Yu-Chiang Frank Wang, Kwang-Ting Cheng, Min-Hung Chen: DoRA: Weight-Decomposed Low-Rank Adaptation. ICML 2024

**Questions:**

1. In Figure 1, which weight types are adapted for each method?
2. For Natural Language Generation tasks, why is SVFT applied to Q,K,V,U,D,O,G for the Gemma family of models, and U,D,O,G for LLaMA-3-8B?
3. For Math and Commonsense reasoning tasks, what are the training, validation and test splits used, and on which split are the hyperparameters tuned?
4. For visual classification tasks, why is LoRA and DoRA applied to all the linear layers instead of Q and V as done in the original VeRA paper [2], which shows better performance at a much lower parameter cost? Is the training data from the same, or different even if the number of samples per class are the same?
    - The performance reported in [2] for VeRA is higher than that reported in this submission (the discrepancy with ViT-B for Food101 is particularly high). What is the cause of this?

**Limitations:**

Authors have addressed the limitations.

---

> ### Author Rebuttal · Authors · 2024-08-07
>
> We sincerely thank the reviewer for the rigorous review and valuable feedback. We have addressed all the weaknesses (W) and questions (Q) with extensive experiments and clarifications.
>
> **W1:** Please refer to our global response **G1** for a comprehensive answer.
>
> **W2:**
> 1, 4. We appreciate you highlighting this oversight. The results marked with * in Table 3 were indeed obtained from BOFT [1].
>
> 2. We apologize for not clearly stating the target modules for baselines in the main paper. This information is provided in Appendix C.5. To clarify:
>    - For Table 1: LoRA and DoRA adapt QKVUD modules as per [2]; BOFT adapts QV modules (adapting more leads to OOM); VeRA adapts GU modules (maximizing trainable parameters within its compatibility constraints).
>    - For Table 3: All methods adapt all linear layers, following [1].
>
> 3. The statistics mentioned in lines 14-16 and 46-48 are extracted from Figure 1 (left) for Gemma-2B on GSM-8K. We will update this in the final version for clarity.
>
> **W3:**\
> 1, 2. We acknowledge that while SVFT reduces trainable parameters, it increases overall GPU memory usage compared to LoRA. As the reviewer correctly noted, reducing trainable parameters decreases memory required for gradients, activations, optimizer state, and various buffers. Following the reviewer's suggestion, we used HuggingFace's internal memory profiler to measure actual peak GPU memory usage. We've included these findings, along with the adapted modules for all baselines (see **G2** in global response). In summary, SVFT uses ~1.2x more memory than LoRA but is comparable to or less than DoRA.
>
> **W4/Q2:** We appreciate the reviewer pointing out the inconsistency in adapted modules. For SVFT, we initially adapted more modules (QKVUDOG) compared to baselines (QKVUD) due to the lower total trainable parameters. However, for LLaMA-3-8B, adapting QKVUDOG exceeds A100-80GB GPU memory (true for baselines as well). Based on our ablation study in Section 5.3, which showed UDOG modules contributing most to performance gains, we chose to adapt UDOG while staying within memory limits.
> We agree that this inconsistency could make it challenging to attribute gains specifically to SVFT. Following the reviewer's advice, we re-ran experiments with SVFT adapting only QKVUD, matching the baselines. We've included these new results in a table below, demonstrating that SVFT maintains similar performance gains even in this consistent setting.
>
> ---
> **Gemma-2B**
> ---
>
> | Method | #Params |Modules| GSM-8K | MATH |
> |:-:|:-:|-|:-:|:-:|
> | LoRA$_{r=4}$|3.28M|Q,K,V,U,D|40.6|14.5|
> | DoRA$_{r=4}$|3.66M|Q,K,V,U,D|41.84|15.04|
> | LoRA$_{r=32}$|26.2M|Q,K,V,U,D|43.06|15.50|
> | DoRA$_{r=16}$|13.5M|Q,K,V,U,D|44.27|16.18|
> | SVFT$^R_{d=12}$|2.98M|Q,K,V,U,D|47.41|**16.72**|
> | SVFT$^R_{d=8}$|3.28M|Q,K,V,U,D,O,G|**47.76**|15.98|
>
> ---
> **LLaMA-3-8B**
> ---
> | Method | #Params | Modules | GSM-8K | MATH |
> |:-:|:-:|:-:|:-:|:-:|
> | LoRA$_{r=4}$|7.07M|Q,K,V,U,D|69.37|22.90|
> | DoRA$_{r=4}$| 7.86M|Q,K,V,U,D|74.37|24.1|
> | LoRA$_{r=32}$|56.6M|Q,K,V,U,D|**75.89**|24.74|
> | DoRA$_{r=16}$|29.1M|Q,K,V,U,D|75.66|24.72|
> | SVFT$^R_{d=24}$|15.98M|Q,K,V,U,D|75.32|**25.08**|
> | SVFT$^R_{d=16}$|17.26M|U,D,O,G|75.43|24.44|
> ---
>
> **W5/Q4:**
>
> **1.** We selected different datasets for ViT-B and ViT-L in Table 4 for two primary reasons:
>    i) To provide a balanced evaluation, we ran ViT-B on relatively simpler datasets (CIFAR100, Flowers102), while testing ViT-L on more complex ones (Food101 and Resisc45).
>    ii) To enhance the table's readability and prevent it from becoming overly large. The complete version of this table, including all dataset-model combinations, is available in Table 9 of the appendix.
>
> **2.** We acknowledge the reviewer's observation that SVFT variants may not consistently outperform LoRA/DoRA across all benchmarks, as evidenced by the Resisc-45 results with ViT-B. However, it's important to note that these results are still comparable to full fine-tuning outcomes. We will update the main paper to reflect this nuanced finding. Additionally, it's worth considering that in vision benchmarks, the classifier head is fully tunable, which may increase the potential for overfitting.
>
> **3, 4.** To clarify our methodology:
>    i) We adopted the target modules for adaptation from [1] and the data split setup from [2]. While [1] uses VTAB-1K for vision experiments, they don't release data splits or sources. Upon careful examination of [3], we discovered that unlike other datasets where evaluation is done on the test split, [3] evaluates Resisc45 on all remaining samples. For consistency, we evaluate on the test split for all vision datasets in our study.
>    ii) To address the second point, we have re-run all vision experiments, adapting only the QV modules – please refer to **G3** in global response for table, which we will include in the revised paper. In short, SVFT offers competitive performance. Adapting QV modules for these vision tasks seems sufficient for all baselines.
>
> **Discrepancy in Food101:** Table 5 in [3] shows ViT-B significantly outperforming ViT-L on Food101, contrary to trends in other datasets. We were unable to reproduce these results for ViT-B, even with full fine-tuning.
>
> **Q1:** See Appendix C.5 L451, or W2.2\
> **Q3:** For Math reasoning, we fine-tune on the MetaMathQA-40K (L181) and evaluate on GSM-8K, MATH following [1]. For Commonsense reasoning, We follow the setting outlined in prior work [2] i.e. all the training splits from the respective datasets are amalgamated into the training set (L184). For baselines, we use the recommended hyperparams from the respective papers, and for SVFT, we do not perform hyperparam tuning and select params outlined in Appendix C.5.
>
> **References** \
> [1] Liu et al. Parameter-efficient orthogonal finetuning via butterfly factorization. ICLR 2024. \
> [2] Liu et al. Dora: Weight-decomposed low-rank adaptation. ICML 2024. \
> [3] Kopiczko et. al: VeRA: Vector-based Random Matrix Adaptation. ICLR 2024

---

> > ### Comment · Reviewer_DEin · 2024-08-07
> >
> > Thank you for your detailed response. Most of my concerns have been addressed. Please include your findings and reasoning for W4, and the findings in the global response in the revised version. I will raise my score from 4 to 6.

---

> > > ### Author Response · Authors · 2024-08-07
> > >
> > > We thank the reviewer for their positive feedback and for raising their score. We will incorporate the findings and reasoning for W4, along with the global response, in the revised version of our paper.

---

### Official Review · Reviewer_x5z5 · 2024-07-10

**Soundness:** 3
**Presentation:** 3
**Contribution:** 2
**Rating:** 3
**Confidence:** 5

**Summary:**

The authors propose a parameter-efficient fine-tuning method with singular vectors (SVFT). SVFT modifies the parameter matrix based on the structure of the original parameter matrix $\mathbf{W}$. By training the coefficient of a sparse combination of outer products of $\mathbf{W}$'s singular vectors, SVFT recovers up to 96% of full fine-tuning performance while training 0.006 to 0.25% of the parameter.

**Strengths:**

It is a simple approach that should be easy to reproduce.

**Weaknesses:**

Overstatement: the authors fine-tune the coefficient of the original parameter matrix $\mathbf{W}$'s singular vectors to achieve that fine-grained control over expressivity, which is good - but practically this is still similar to the existing methods. Stating in the abstract that \`\` We propose SVFT, a simple approach that fundamentally differs from existing methods: the structure imposed on $\Delta \mathbf{W}$ depends on the specific weight matrix $\mathbf{W}$ .\'\' suggests more extensive comparison with existing methods as well.

[1] Peng, Zelin, et al. SAM-PARSER: Fine-Tuning SAM Efficiently by Parameter Space Reconstruction, AAAI 2024.

[2] Han, Ligong, et al. SVDiff: Compact Parameter Space for Diffusion Fine-Tuning, ICCV 2023.

I could not find the comparison of $\text{SVFT}_P$, $\text{SVFT}_B$, $\text{SVFT}^R_d$ and $\text{SVFT}^T_d$ in Tables 1-4. At the moment it's unclear how to choose the sparsity pattern $\Omega$.

**Questions:**

if $\text{SVFT}^{R}_{d=2}$, $k=?$

if $\text{SVFT}^{T}_{d=2}$, $k=?$

What is the purpose of proposing $\text{SVFT}^T_d$ if it has not shown any performance advantage in any experiment?

How to choose the appropriate sparsity pattern $\Omega$?

For model performance, does randomness in $\text{SVFT}^T_d$ lead to instability?

**Limitations:**

Yes.

---

> ### Author Rebuttal · Authors · 2024-08-06
>
> We thank the reviewer for their valuable feedback. We address the weaknesses (W) and questions (Q) raised by the reviewer below.
>
> **W1:** We thank the reviewer for pointing us to these papers (SAM-PARSER [1], SVDiff [2]). Indeed we are not the first one to explore the singular vectors/values space. [1] and [2] are essentially equivalent to our SVFT$^P$ variant, but we note that our main insight and contribution is that significant gains can be achieved by learning *off-diagonal* interactions in $M$.
>
> Doing our due diligence, in the following Table, we compare SVFT$^P$ against SVF (which is also equivalent to SAM-PARSER and SVDiff) on GSM-8K and Math benchmarks. Results in the Table below show SVF performs similarly to SVFT$^P$ and worse than SVFT$^R$ as we had expected (the same holds true for SVFT$^B$ and SVFT$^T$ from the table in W2). We will acknowledge these papers and add a discussion in our final version of the paper.
>
> | Baseline | #Trainable Params | Target Modules | GSM-8K | MATH |
> |:----------:|:---------:|:----------------:|:-------:|:------:|
> | SVF | 120K | Q, K, V, U, D | 36.39 | 13.96 |
> | SVFT$^P$ | 120K | Q, K, V, U, D | 36.39 | 13.86 |
> | SVF | 194K | Q, K, V, U, D, O, G | 39.12 | 14.02 |
> | SVFT$^P$ | 194K | Q, K, V, U, D, O, G | 40.34 | 14.38 |
> | SVFT$^R_{d=16}$ | 6.35M | Q, K, V, U, D, O, G| **50.03** | **15.56** |
>
> To clarify our contribution, we will revise the highlighted statement to: \
> *SVFT enables a trade-off between the number of trainable parameters and model expressivity by allowing a flexible number of off-diagonal interactions between singular vectors in $\Delta$W, distinguishing it from previous SVD-based methods.*
>
> **W2/Q3:** We note that simple heuristics for sparsity patterns in $M$ significantly improve performance over previous PEFT techniques. While most variants (except SVFT$^P$) offer comparable performance, the banded pattern (SVFT$^B$) slightly outperforms others. We hypothesize that a more principled approach to finding optimal sparsity patterns for specific tasks and base models may yield further performance gains. We leave it as future work to explore task-specific and model-specific sparsity patterns. We will revise our paper to reflect these new results and findings.
>
> | Results on fine-tuning with SVFT using different $M$ parameterizations. |
> |:---|
>
> | Structure |    | Gemma-2B |   |  | Gemma-7B |  |  | LLaMA-3-8B |  |
> |:----------|:-------:|:-------:|:-------:|:-------:|:-------:|:-------:|:--------:|:-------:|:-------:|
> |           | #Trainable Params | GSM-8K | MATH | #Trainable Params | GSM-8K | MATH | #Trainable Params | GSM-8K | MATH |
> | Plain     | 0.2M    | 40.34  | 14.38 | 0.43M   | 73.50  | 27.30 | 0.48M   | 69.22  | 20.44 |
> | Banded    | 6.4M    | 47.84  | **15.68** | 19.8M   | **76.81**  | **29.98** | 17.2M   | **75.43**  | **24.44** |
> | Random    | 6.4M    | **50.03**  | 15.56 | 19.8M   | 76.35  | 29.86 | 17.2M   | 74.07  | 23.78 |
> | Top-k     | 6.4M    | 49.65  | 15.32 | 19.8M   | 76.34  | 29.72 | 17.2M   | 73.69  | 23.96 |
>
> **Q1:** For SVFT$^R$ and SVFT$^T$ variants, we select the number of trainable parameters in each target module's weight matrix to match SVFT$^B$'s parameter count for a given 'k'. This is calculated as $k=n(2d+1) - d(d+1)$, where $d$ is the number of off-diagonals in the banded matrix and $n$ is the matrix's smaller dimension. This is to have a fair comparison between methods.  Eg: For $Q$ matrix on Gemma-2B ($n$=2048), when off-diagonals $d=2$, then $k=10,234$ for both SVFT$^T_{d=2}$ and SVFT$^R_{d=2}$ which is the same as that of SVFT$^B_{d=2}$. In practice, ‘$k$’ can be set to any value for these two variants.
>
> **Q2:** We propose SVFT$^T$ to explore a broad spectrum of possible selection criteria of $M$’s sparsity pattern. Our experiments show that simple heuristics-based sparsity patterns can improve performance. In future work, we will explore more principled approaches for finding the optimal pattern.
>
> **Q4:** We did not observe any instability in our experiments. In the worst-case scenario, SVFT$^T$ will be equivalent to SVFT$^R$, which performs well in our experiments.
>
> **References**\
> [1] Peng, Zelin, et al. SAM-PARSER: Fine-Tuning SAM Efficiently by Parameter Space Reconstruction, AAAI 2024.\
> [2] Han, Ligong, et al. SVDiff: Compact Parameter Space for Diffusion Fine-Tuning, ICCV 2023.

---

> ### Comment · Reviewer_x5z5 · 2024-08-08
>
> 1. For $\text{SVFT}^{\mathbf{R}}$ and $\text{SVFT}^{\mathbf{T}}$, $d$ is a meaningless notation. $\text{SVFT}^{\mathbf{R}}_d$ and $\text{SVFT}^{\mathbf{T}}_d$ will be misleading to the reader.
>
> 2. As a parameter-efficient fine-tuning method, Why not compare VeRA, LoRA, DoRA, BOFT, and SVFT with the same (or similar) number of parameters in Tables 1-4? What are the criteria for evaluating the performance of PEFT methods?
>
> 3. In both the original and newly provided comparison results, $\textbf{SVFT}^{T}$ is not the best-performing performance. Why is $\textbf{SVFT}^{T}$ proposed to explore the broad spectrum of possible selection criteria of $\mathbf{M}$`s sparsity pattern?
>
> 4. If the main insight and contribution is that significant gains can be achieved by learning off-diagonal interactions in $\mathbf{M}$, a comparison of Random and Top-k with Plain in the same parameter quantities must be provided.
>
> 5. It is worth noting that the singular value space is not the only avenue to explore the effects of off-diagonal interactions in $\mathbf{M}$. Like OFT[1] and BOFT[2] which introduced in Related Work section, $\mathbf{h} = (\mathbf{M} \mathbf{W_0}) \mathbf{x}$ or $\mathbf{h} = (\mathbf{W_0} \mathbf{M}) \mathbf{x}$ can also learn the off-diagonal interactions in $\mathbf{M}$, and the four sparsity patterns of ``Random, Top-k, Plain and Banded‘’ are also applicable choices in this situation. As a reparameterization-based method, why can not the sparse trainable matrix $\mathbf{M}$ directly update the pre-trained weight matrix $\mathbf{W}_0$? Why are the Singular Vectors a better way to guide fine-tuning? Please provide detailed analysis and comparison results to enhance the insight and contribution of this work.
>
> [1] Controlling text-to-image diffusion by orthogonal fine-tuning. NIPS23.
> [2] Parameter-efficient orthogonal finetuning via butterfly factorization. ICLR24.

---

> ### Author Response · Authors · 2024-08-10
> **Response 1/2**
>
> We thank the reviewer for their prompt response and new feedback. We address their comments (C) below. Please see *Response 2/2* for **C4, C5**.
>
> **C1.** We agree that using $d$ for Top-k and Random methods is redundant and potentially confusing. We initially aimed for consistency across all methods, including $SVFT^B$, to facilitate comparisons. In the final version, we will: 1. Explicitly state the number of trainable parameters in notation for Top-k and Random. 2. Clearly differentiate notation between banded and non-banded methods. 3. Briefly explain this change in notation. 4. Update all relevant tables, figures, and discussions accordingly.
>
> **C2.** We indeed compared VeRA, LoRA, DoRA, BOFT, and SVFT with similar parameter counts as shown in Figures 1 and 5. Note we cannot precisely match the number of parameters because, for some baselines, adjustments can only be made in increments greater than 1, e.g increasing rank by 1 introduces 2d parameters.
>
> Method-specific constraints limited parameter adjustments:
>  - VeRA: This method shares frozen adapters across compatible layers, making it difficult to significantly increase the total trainable parameters by simply increasing intermediate dimensions.
>  - BOFT: Adapting any additional module beyond QV leads to memory issues, as the pre-multiplication matrix R is a product of multiple orthogonal matrices. This limitation is also noted in Section 7 of the BOFT paper.
>
> For reviewer’s convenience, We provide tables with clear demarcation for easier comparison. SVFT variants show notable gains with comparable parameter counts.
>
> | Model Name         |  | Gemma-2B         | |
> | ------------------ | ---------------- | ----- | ----- |
> | Fine Tuning Method | Trainable Params | GSM8K  | MATH |
> | VeRA$_{r=1024}$        | 627K             | 36.77 | 14.12 |
> | LoRA$_{r=1}$           | 820K             | 32.97 | 13.04 |
> | SVFT$^B_{d=2}$       | 967K             | **44.65** | **14.48** |
> | DoRA$_{r=1}$           | 1.198M           | 35.25 | 13.04 |
> | BOFT$^{m=2}_{b=8}$       | 1.221M           | 36.01 | 12.13 |
>
> | Model Name         |  | Gemma-7B         | |
> | ------------------ | ---------------- | ---- | ---- |
> | Fine Tuning Method | Trainable Params | GSM8K | MATH |
> | VeRA$_{r=1024}$        | 430K             | 71.11 | 27.04 |
> | SVFT$^P$             | 429K             | **73.5** | **27.3** |
> | LoRA$_{r=1}$           | 820K             | 72.4 | 26.28 |
>
> | Model Name       |  | Llama-3-8B       | |
> | ------------------ | ---------------- | --- | --- |
> | Fine Tuning Method | Trainable Params | GSM8K | MATH |
> | SVFT$^P$             | 483K             | **69.22** | **20.44** |
> | VeRA$_{r=1024}$        | 983K             | 63.76 | 20.28 |
>
> | Model Name       |  | Llama-3-8B       | |
> | ------------------ | ---------------- | --- | --- |
> | Fine Tuning Method | Trainable Params | GSM8K | MATH |
> | LoRA$_{r=1}$           | 1.770M           | 68.84 | 20.94 |
> | DoRA$_{r=1}$           | 2.556M           | 68.3 | 21.96 |
> | SVFT$^B_{d=2}$         | 3.603M           | **71.42** | **22.72** |
> | BOFT$^{m=2}_{b=8}$       | 4.358M           | 67.09 | 21.64 |
>
> For criteria for evaluating PEFT methods, we considered several factors:
> 1. Performance: How well the method improves the model's performance on the target task (Tables 1-4).
> 2. Parameter Efficiency: The ratio of performance improvement to the number of added parameters (Figure 1 and 5).
> 3. Memory Efficiency: The method's ability to optimize memory usage during training and inference (Please see our global response).
> 4. Scalability: How well the method performs across different model sizes and task types (Table 1).
>
> **C3** In our study, we investigate various simple sparsity patterns, including $SVFT^T$, and analyze their impact on performance. This analysis serves as an ablative study to understand the effectiveness of different sparsity patterns. Considering reviewer's feedback, our final paper will highlight that other sparsity patterns outperform the $SVFT^T$.

---

> ### Author Response · Authors · 2024-08-10
> **Response 2/2**
>
> **C4.** Comparing $SVFT^R$ and $SVFT^T$ with $SVFT^P$ using the same number of parameters isn't very meaningful. $SVFT^P$ applies a *constrained* full-rank update. For $SVFT^R$ and $SVFT^T$ , when the total number of trainable parameters is equal to that in $SVFT^P$ (i.e., the total elements in the main diagonal of Σ for a chosen W), the rank of the perturbation is reduced compared to adapting the entire main diagonal (Rank Lemma in Sec 3.2). We introduced SVFT variants to offer flexibility in the total number of trainable parameters. Significant gains are observed when the total number of learnable parameters is greater than in Plain, starting notably at d=2. In line 162, from equation (2), in our paper, $u_i$​ appears in multiple summation terms in SVFT, which is less likely when only learning main diagonal number of parameters in W. To substantiate our point, please see below table -- we only report the rank of $\Delta W$ for Q for clarity. However, similar findings hold across remaining target modules.
>
> | Method | #Params | Rank of $\Delta W_Q$ | GSM-8K | MATH |
> | :--: | :--: | :--: | :--: | :--: |
> | SVFT$^P$ | 194K | **2048** | 40.34 | 14.38 |
> | SVFT$^R$ | 194K | 1109 | 39.65 | 13.98 |
> | SVFT$^R$ | 967K | **2020** | **44.65** | **14.48** |
>
>
> **C5.**
> - (i) We agree that there are other avenues worth exploring regarding the effects of the off-diagonal interactions we introduced in M. However, OFT and BOFT impose specific structures on the pre/post multiplication matrix $R$ to preserve orthogonality and the spectral norm of $W$. Applying our sparsity patterns to $R$ would violate these conditions, potentially leading to training instability (see the spectral properties discussed in Section 5 of BOFT).
>
> - (ii) Fine-tuning with fixed sparse masks has been studied before and compared to early PEFT methods like Bit-Fit [2] and Adapter Tuning [3]. For example, [1] demonstrates that directly updating $W_0$ with a sparse trainable random matrix $M$ is inferior to Bit-Fit. Recent PEFT methods, such as LoRA and VeRA, show significant improvements over Bit-Fit. We compare our approach against LoRA and VeRA in our experiments.
>
> Our work is largely empirical with a few theoretical insights. The question of why Singular Vectors provide a better way to guide fine-tuning is intriguing and could be a potential direction for future theoretical study.
>
> [1] Training Neural Networks with Fixed Sparse Masks. NeurIPS 2021.\
> [2] BitFit: Simple Parameter-efficient Fine-tuning for Transformer-based Masked Language-models. ACL 2022\
> [3] Parameter-Efficient Transfer Learning for NLP. ICML 2019

---

> > ### Comment · Reviewer_x5z5 · 2024-08-12
> >
> > Thank you for your response. Based on the above discussion, I think the exploration of this work is neither comprehensive nor thorough. It is quite obvious that increasing the number of trainable parameters in the singular value space can improve PEFT performance. The author does not clearly explain how to design an appropriate sparse pattern based on the characteristics of downstream tasks. The author suggests further exploration of the \textbf{SVFT}$^T$, which is weaker than the others, but the significance and value of this direction are not clearly demonstrated in this work. Taking all the above factors into consideration, I give the final vote.

---

> ### Author Response · Authors · 2024-08-13
>
> We sincerely thank the reviewer for their feedback and for actively engaging in the discussion. We address their concerns below:
>
> > **"The exploration of this work is neither comprehensive nor thorough."**
>
> We respectfully disagree with this assessment. Our work includes experiments on 22 datasets across both language and vision tasks, involving 4 different language models and 5 baselines, with various configurations to match total trainable parameters. Moreover, during the rebuttal phase, we provided additional analyses, including memory usage and an ablation study on different sparsity patterns. If there are specific areas where the reviewer feels our exploration is lacking, we would appreciate further clarification.
>
>
> > **“The author does not clearly explain how to design an appropriate sparse pattern based on the characteristics of downstream tasks."**
>
> Our primary insight focuses on the inclusion of *additional* off-diagonal elements, with empirical evidence showing that even simple sparse patterns can outperform previous PEFT methods across different tasks. While designing task-specific sparse patterns is indeed valuable, it is beyond the scope of this work.
>
> > **"The author suggests further exploration of SVFT$^T$…"**
>
> We believe there may be a misunderstanding here. As clarified in our previous responses, our analysis was intended as an ablation study to evaluate the effectiveness of different sparsity patterns.
>
> Throughout the rebuttal, we have addressed several of the previously raised concerns with new experiments and referenced relevant published works to provide additional context. We kindly request the reviewer to reconsider their evaluation of our work in light of these responses.

---

### Official Review · Reviewer_BEJi · 2024-07-10

**Soundness:** 3
**Presentation:** 3
**Contribution:** 3
**Rating:** 5
**Confidence:** 5

**Summary:**

This paper presents a new PEFT method to enhance the fine-tuning efficiency of large language models (LLMs). The approach uses SVD to decompose pre-trained weights, initializing adapters with eigenvectors and eigenvalues, and adjusts within the eigenvalue space. Extensive experiments show that the proposed method requires fewer parameters while achieving performance that matches or exceeds other PEFT methods.

**Strengths:**

The paper is well-written and easy follow. The motivation is clear, the method is straightforward, and it has been evaluated on mainstream tasks in both NLP and CV fields.

**Weaknesses:**

1. Introducing the M matrix indeed allows for more ways to modify eigenvalues, but the paper does not discuss the differences and advantages between directly fine-tuning the SVD eigenvalue space and the proposed method.
2. Decomposing the pre-trained weights directly results in a significant increase in the number of parameters during training, leading to much higher training costs compared to methods like LoRA. This is the main drawback of fine-tuning the eigenvalue space.
3. The approach of fine-tuning the singular value space after SVD decomposition of pre-trained weights is not novel. The paper lacks a summary of such methods, such as [1] Singular value fine-tuning: Few-shot segmentation requires few-parameters fine-tuning and [2] Svdiff: Compact parameter space for diffusion fine-tuning.

**Questions:**

same as weaknesses

**Limitations:**

This paper does not reflect any potential negative societal impact.

---

> ### Author Rebuttal · Authors · 2024-08-06
>
> We thank the reviewer for their valuable feedback. We address the weaknesses (W) raised by the reviewer below.
>
> **W1:** We emphasize that our main contribution is making off-diagonal elements in $M$ learnable; this allows us to smoothly trade-off between the number of trainable parameters and expressivity (which is not possible by only fine-tuning the singular values -- the main diagonal of $M$). We justify this theoretically (Expressivity Lemma in Section 3.2) and empirically (in all our results SVFT$^B$, SVFT$^R$, SVFT$^T$ outperforms SVFT$^P$).
>
> **W2:** We acknowledge that while SVFT reduces trainable parameters, it increases overall GPU memory usage compared to LoRA.  However, reducing trainable parameters decreases the memory required for gradients, activations, optimizer state, and various buffers. To address the reviewer's concern, we used HuggingFace's internal memory profiler to log actual peak GPU memory usage. We've included these findings, along with the adapted modules for all baselines, in the table below. In summary, SVFT outperforms LoRA while using ~1.2x more memory. It is worth noting that SVFT's GPU memory usage is either similar to or lower than that of DoRA.
>
> ---
>
> ### **Gemma-2B**
>
> | Method       | Target Modules | #Trainable Params | GPU Mem (GB)  | GSM-8K | MATH  |
> |:-:|:-:|:-:|:-:|:-:|:-:|
> | LoRA$_{r=4}$      | Q, K, V, U, D   | 3.28M  | 18.8839 | 40.63 | 14.5  |
> | DoRA$_{r=4}$      | Q, K, V, U, D          | 3.66M  | 24.5798 | 41.84 | 15.04 |
> | LoRA$_{r=32}$     | Q, K, V, U, D          | 26.2M             | 19.0558 | 43.06 | 15.5  |
> | DoRA$_{r=16}$     | Q, K, V, U, D          | 13.5M             | 24.6430 | 44.27 | 16.18 |
> | SVFT$^R_{d=12}$  | Q, K, V, U, D          | 2.98M             | 20.4988 | 47.61 | **16.72** |
> | SVFT$^P$       | Q, K, V, U, D, O, G        | 194K              | 21.8984 | 40.34 | 14.38 |
> | SVFT$^R_{d=8}$   | Q, K, V, U, D, O, G         | 3.28M             | 22.0224 | 47.76 | 15.98 |
> | SVFT$^R_{d=16}$ | Q, K, V, U, D, O, G         | 6.35M             | 22.1465 | **50.03** | 15.56 |
>
> ---
>
> ### **Gemma-7B**
>
> | Method      | Target Modules | #Trainable Params | GPU Mem (GB)  | GSM-8K | MATH  |
> |:-:|:-:|:-:|:-:|:-:|:-:|
> | LoRA$_{r=4}$     | Q, K, V, U, D          | 8.60M  | 63.5711 | 73.76 | 28.3  |
> | DoRA$_{r=4}$     | Q, K, V, U, D         | 9.72M             | 78.7007 | 75.43 | 28.46 |
> | LoRA$_{r=32}$    | Q, K, V, U, D        | 68.8M   | 64.2403 | 76.57 | 29.34 |
> | DoRA$_{r=16}$    | Q, K, V, U, D          | 35.5M  | 78.9913 | 74.52 | 29.84 |
> | SVFT$^P$      | Q, K, V, U, D, O, G         | 429K | 76.2630 | 73.5  | 27.3  |
> | SVFT$^R_{d=8}$  | Q, K, V, U, D, O, G        | 9.80M   | 76.6506 | 73.62 | 28.36 |
> | SVFT$^R_{d=16}$ | Q, K, V, U, D, O, G         | 19.8M    | 77.0363 | **76.81** | **29.98** |
> ---
>
> ### **Llama-3-8B**
>
> | Method      | Target Modules | #Trainable Params | GPU Mem (GB)   | GSM-8K | MATH  |
> |:-:|:-:|:-:|:-:|:-:|:-:|
> | LoRA$_{r=1}$     | Q, K, V, U, D  | 1.77M | 57.8217 | 68.84 | 20.94 |
> | DoRA$_{r=1}$     | Q, K, V, U, D | 2.56M  | 70.1676 | 68.30 | 21.96 |
> | LoRA$_{r=32}$    | Q, K, V, U, D | 56.6M  | 58.4089 | 75.89 | 24.74 |
> | DoRA$_{r=16}$    | Q, K, V, U, D | 29.1M  | 70.4383 | 75.66 | 24.72 |
> | SVFT$^B_{d=24}$ | Q, K, V, U, D | 15.98M | 71.5224 | 75.32 | **25.08** |
> | SVFT$^R_{d=12}$ | U, D, O, G | 13.1M | 70.3681 | **75.90** | 24.22 |
> | SVFT$^P$      | Q, K, V, U, D, O, G    | 483K  | 73.9495 | 69.22 | 20.44 |
> | SVFT$^R_{d=12}$ | Q, K, V, U, D, O, G | 15.9M | 71.5224 | 73.99 | **25.08** |
> ---
>
>
> **W3:** We thank the reviewer for pointing us to these papers (SVF [1], SVDiff [2]). Indeed [1] and [2] are essentially equivalent to SVFT$^P$, but we note that our main insight and contribution is that significant gains can be achieved by learning *off-diagonal* interactions in $M$.
>
> Doing our due diligence, in the following Table, we compare SVFT$^P$ against SVF on GSM-8K and Math benchmarks. Results in the Table below show SVF performs similar to SVFT$^P$ and worse than SVFT$^{R}$ as we had expected. We will acknowledge these papers and add a discussion in our final version of the paper.
>
> ---
> | Baseline | #Trainable Params | Target Modules | GSM-8K | MATH |
> |:----------:|:---------:|:----------------:|:-------:|:------:|
> | SVF | 120K | Q, K, V, U, D | 36.39 | 13.96 |
> | SVFT$^P$ | 120K | Q, K, V, U, D | 36.39 | 13.86 |
> | SVF | 194K | Q, K, V, U, D, O, G | 39.12 | 14.02 |
> | SVFT$^P$ | 194K | Q, K, V, U, D, O, G | 40.34 | 14.38 |
> | SVFT$^B_{d=16}$ | 6.35M | Q, K, V, U, D, O, G | 47.84 | **15.68** |
> | SVFT$^R_{d=16}$ | 6.35M | Q, K, V, U, D, O, G | **50.03** | 15.56 |
> | SVFT$^T_{d=16}$ | 6.35M | Q, K, V, U, D, O, G | 49.65 | 15.32 |
> ---
>
>
>
> **References**\
> [1] Sun et al.: Singular Value Fine-tuning: Few-shot Segmentation requires Few-parameters Fine-tuning. NeurIPS 2022\
> [2] Han et al. SVDiff: Compact Parameter Space for Diffusion Fine-Tuning, ICCV 2023.

---

> > ### Comment · Reviewer_BEJi · 2024-08-12
> >
> > Thank you for your feedback; it addressed my concerns. I will raise my score to 5. However, I still believe that the SVFT paradigm has already been widely discussed and used, which makes the motivation and contribution in the paper unclear. I hope this can be clarified in the final version.

---

> > > ### Comment · Area_Chair_YgWR · 2024-08-12
> > > **Request your clarifications**
> > >
> > > Dear Reviewer,
> > >
> > > Thank you for your comments and efforts.
> > >
> > > I wonder if you could elaborate on "I still believe that the SVFT paradigm has already been widely discussed and used, which makes the motivation and contribution in the paper unclear.", since this could provide us better understanding of the novelty of the work from your perspective.
> > >
> > > Thanks

---

### Official Review · Reviewer_Cyjn · 2024-07-11

**Soundness:** 3
**Presentation:** 3
**Contribution:** 3
**Rating:** 7
**Confidence:** 4

**Summary:**

This paper proposes a parameter-efficient fine-tuning (PEFT) method named SVFT. Instead of imposing learnable parameters directly on the pre-trained weight matrices, it applies singular value decomposition to the pre-trained weight matrices and then tunes only the coefficients of the singular values. The method is validated on various vision/language tasks, showing comparable or better performance compared to multiple existing PEFT methods.

**Strengths:**

* The paper is well-written and easy to follow
* The method is novel. Studying the behavior of SVFT in more depth can be beneficial to the PEFT community.
* On both language and vision tasks, SVFT shows comparable or better performance than LoRA, DoRA, VERA, and BOFT, demonstrating the efficacy of the method.

**Weaknesses:**

* The comparison between VeRA and $\text{SVFT}^P$ is interesting. Design-wise, they are quite similar, yet $\text{SVFT}^P$ leads to much better performance than VeRA in most cases, even though VeRA has a larger intermediate dimension (e.g., Table 2). I wonder if the authors can elaborate more on this matter.
* In many cases, $d=2$ seems to be sufficient for $\text{SVFT}^B$. However, increasing $d$ for $\text{SVFT}^B$ does not necessarily improve performance (e.g., Table 4 in the main paper or Table 9 in the appendix). How should we interpret this, and how should we select the suitable $d$ in practice?
* What advice do the authors have for choosing between $\text{SVFT}^B$ and $\text{SVFT}^R$? They seem to have comparable performance on different benchmarks.

**Questions:**

See Weaknesses

---

> ### Author Rebuttal · Authors · 2024-08-06
>
> We thank the reviewer for their valuable feedback. We address the weaknesses (W) raised by the reviewer below.
>
> **W1:** Recall that VeRA injects frozen low-rank random matrices with learnable scaling vectors resulting in weight updates independent of the pre-trained weights. SVFT$^P$ derives its weight updates directly from the pre-trained matrix $W$. \
> VeRA differs from SVFT$^P$ in two ways: \
> i) VeRA achieves high parameter efficiency by sharing frozen adapters across compatible layers, a choice that no other previous methods take (e.g., LoRA, DoRA). This particular design choice gives it lower expressiveness than what may have been possible for other methods with the same intermediate dimension. \
> ii) The perturbation directions that VeRA operates on (i.e., its frozen adapters) are chosen independent of the underlying matrix that it is trying to perturb.
>
> We note that despite the sharing of parameters across layers in VeRA, it ends up having more trainable parameters per layer compared to SVFT$^P$. For example, in Table 2, we achieve better results with SVFT$^P$ using dims=4096 (0.51M params) compared to VeRA with dims=2048 (1.49M params). SVFT$^P$'s use of singular vectors gives it layer-wise specialization.
>
>
> **W2:** As '$d$' (off-diagonals) (or '$r$' for other PEFT methods) increases, performance typically improves. However, overfitting can occur with any method, especially with smaller datasets or models. Figure 5 (left) in the paper shows $DoRA_{r=16}$ underperforming DoRA$_{r=4}$ on language tasks. In Tables 4 and 9, the additional fine-tuning of the classifier head may contribute to overfitting.
>
>
> **W3:** We note that simple heuristics for sparsity patterns in $M$ significantly improve performance over previous PEFT techniques. While most variants (except SVFT$^P$) offer comparable performance, the banded pattern (SVFT$^B$) slightly outperforms others. We hypothesize that a more principled approach to finding optimal sparsity patterns for specific tasks and base models may yield further performance gains. We leave it as future work to explore task-specific and model-specific sparsity patterns. We will revise our paper to reflect these new results and findings.
>
>
> | Results on fine-tuning with SVFT using different $M$ parameterizations. |
> |:---|
>
> | Structure |    | Gemma-2B |   |  | Gemma-7B |  |  | LLaMA-3-8B |  |
> |:----------|:-------:|:-------:|:-------:|:-------:|:-------:|:-------:|:--------:|:-------:|:-------:|
> |           | #Trainable Params | GSM-8K | MATH | #Trainable Params | GSM-8K | MATH | #Trainable Params | GSM-8K | MATH |
> | Plain     | 0.2M    | 40.34  | 14.38 | 0.43M   | 73.50  | 27.30 | 0.48M   | 69.22  | 20.44 |
> | Banded    | 6.4M    | 47.84  | **15.68** | 19.8M   | **76.81**  | **29.98** | 17.2M   | **75.43**  | **24.44** |
> | Random    | 6.4M    | **50.03**  | 15.56 | 19.8M   | 76.35  | 29.86 | 17.2M   | 74.07  | 23.78 |
> | Top-k     | 6.4M    | 49.65  | 15.32 | 19.8M   | 76.34  | 29.72 | 17.2M   | 73.69  | 23.96 |

---

> > ### Comment · Reviewer_Cyjn · 2024-08-10
> >
> > Thank you for the detailed response and extensive results. I understand that selecting optimal sparsity patterns requires further effort, and I encourage the authors to explore this direction. Aside from that, I am satisfied with the authors' response and will maintain my original positive rating of acceptance.

---

> > > ### Author Response · Authors · 2024-08-11
> > >
> > > Thank you for the positive feedback and encouragement. We're glad our response addressed your concerns, and we appreciate your continued support.

---

### Author Rebuttal · Authors · 2024-08-06

We thank all reviewers for their valuable feedback. We address overlapping concerns below.

---

**G1: Missing Discussion of Related Work.**\
SVF, SVDiff, SAM-PARSER are equivalent to SVFT$^{P}$ barring minor implementation differences. We compare these methods against SVFT$^{P}$ on GSM-8K and Math benchmarks. Results are shown in the table below and observe similar performance. For completeness, we will revise our paper to include these results.

---

### **SVF v/s SVFT$^{P}$ on Gemma-2B**
---
| Baseline | #Trainable Params | Target Modules | GSM-8K | MATH |
|:-:|:-:|:-:|:-:|:-:|
| SVF | 120K | Q, K, V, U, D | 36.39 | 13.96 |
| SVFT$^P$ | 120K | Q, K, V, U, D | 36.39 | 13.86 |
| SVF | 194K | Q, K, V, U, D, O, G | 39.12 | 14.02 |
| SVFT$^P$ | 194K | Q, K, V, U, D, O, G | 40.34 | 14.38 |
| SVFT$^R_{d=16}$ | 6.35M | Q, K, V, U, D, O, G| **50.03** | **15.56** |
---

**G2: Memory usage during Training.**\
We acknowledge that while SVFT reduces trainable parameters, it increases overall GPU memory usage compared to LoRA. However, reducing trainable parameters decreases the memory required for gradients, activations, optimizer state, and various buffers. To substantiate this, we used HuggingFace's internal memory profiler to measure peak GPU memory usage. We have included these findings along with the adapted modules for all baselines in the table below. In summary, SVFT uses ~1.2x more memory than LoRA but is comparable to or less than DoRA. Also note that since SVFT requires less trainable parameters than other methods, the cost of storing U and V can be amortized, and can eventually be cheaper than LoRA/DoRA, especially when the number of parallel adaptations is in millions [1].

---

### **Gemma-2B**

---

| Method       | Target Modules | #Trainable Params | GPU Mem (GB)  | GSM-8K | MATH  |
|:-:|:-:|:-:|:-:|:-:|:-:|
| LoRA$_{r=4}$      | Q, K, V, U, D   | 3.28M  | 18.8839 | 40.63 | 14.5  |
| DoRA$_{r=4}$      | Q, K, V, U, D          | 3.66M  | 24.5798 | 41.84 | 15.04 |
| LoRA$_{r=32}$     | Q, K, V, U, D          | 26.2M             | 19.0558 | 43.06 | 15.5  |
| DoRA$_{r=16}$     | Q, K, V, U, D          | 13.5M             | 24.6430 | 44.27 | 16.18 |
| SVFT$^R_{d=12}$  | Q, K, V, U, D          | 2.98M             | 20.4988 | 47.61 | **16.72** |
| SVFT$^P$       | Q, K, V, U, D, O, G        | 194K              | 21.8984 | 40.34 | 14.38 |
| SVFT$^R_{d=8}$   | Q, K, V, U, D, O, G         | 3.28M             | 22.0224 | 47.76 | 15.98 |
| SVFT$^R_{d=16}$ | Q, K, V, U, D, O, G         | 6.35M             | 22.1465 | **50.03** | 15.56 |

---

### **Gemma-7B**

---

| Method      | Target Modules | #Trainable Params | GPU Mem (GB)  | GSM-8K | MATH  |
|:-:|:-:|:-:|:-:|:-:|:-:|
| LoRA$_{r=4}$     | Q, K, V, U, D          | 8.60M  | 63.5711 | 73.76 | 28.3  |
| DoRA$_{r=4}$     | Q, K, V, U, D         | 9.72M             | 78.7007 | 75.43 | 28.46 |
| LoRA$_{r=32}$    | Q, K, V, U, D        | 68.8M   | 64.2403 | 76.57 | 29.34 |
| DoRA$_{r=16}$    | Q, K, V, U, D          | 35.5M  | 78.9913 | 74.52 | 29.84 |
| SVFT$^P$      | Q, K, V, U, D, O, G         | 429K | 76.2630 | 73.5  | 27.3  |
| SVFT$^R_{d=8}$  | Q, K, V, U, D, O, G        | 9.80M   | 76.6506 | 73.62 | 28.36 |
| SVFT$^R_{d=16}$ | Q, K, V, U, D, O, G         | 19.8M    | 77.0363 | **76.81** | **29.98** |

---

### **Llama-3-8B**

---

| Method      | Target Modules | #Trainable Params | GPU Mem (GB)   | GSM-8K | MATH  |
|:-:|:-:|:-:|:-:|:-:|:-:|
| LoRA$_{r=1}$     | Q, K, V, U, D  | 1.77M | 57.8217 | 68.84 | 20.94 |
| DoRA$_{r=1}$     | Q, K, V, U, D | 2.56M  | 70.1676 | 68.30 | 21.96 |
| LoRA$_{r=32}$    | Q, K, V, U, D | 56.6M  | 58.4089 | 75.89 | 24.74 |
| DoRA$_{r=16}$    | Q, K, V, U, D | 29.1M  | 70.4383 | 75.66 | 24.72 |
| SVFT$^B_{d=24}$ | Q, K, V, U, D | 15.98M | 71.5224 | 75.32 | **25.08** |
| SVFT$^R_{d=12}$ | U, D, O, G | 13.1M | 70.3681 | **75.90** | 24.22 |
| SVFT$^P$      | Q, K, V, U, D, O, G    | 483K  | 73.9495 | 69.22 | 20.44 |
| SVFT$^R_{d=12}$ | Q, K, V, U, D, O, G | 15.9M | 71.5224 | 73.99 | **25.08** |

---

**G3: Adapting QV modules for Vision Benchmarks.**\
Following a similar setup in [2], we have re-run all vision experiments, adapting only the QV modules. In short, SVFT offers competitive performance. Adapting QV modules for these vision tasks seems sufficient for all baselines.

---

| Method        | #Trainable Params | CIFAR100 | Food101 | Flowers102 | RESISC45 |
|:-:|:-:|:-:|:-:|:-:|:-:|
| **ViT Base**  |    |          |         |            |          |
| Head          | -                  | 78.58    | 75.14   | 98.71      | 64.42    |
| Full          | 85.8M         | 85.02    | 75.41   | 99.16      | 75.3   |
| LoRA$_{r=8}$      | 294.9K            | 85.65| 76.13 | 99.14     | 74.01    |
| VeRA$_{r=256}$    | 24.6K             | 84       | 74.02   | 99.1       | 71.86    |
| SVFT$^P$        | 18.5K             | 83.78    | 74.43   | 98.99      |  70.55  |
| SVFT$^R_{d=4}$    | 165.4K            | 84.85    | 76.45   | 99.17      | 74.53    |
| SVFT$^B_{d=4}$    | 165.4K            | 84.65    | 76.51   | 99.21 | 75.12    |
| **ViT Large** |                    |          |         |            |          |
| Head          | -                  | 79.14    | 75.66   | 98.89      | 64.99    |
| Full          | 303.3M        | 87.37    | 78.67   | 98.88      | 80.17|
| LoRA$_{r=8}$      | 786.4K            | 87.36    | 78.95| 99.24  | 79.55    |
| VeRA$_{r=256}$    | 61.4K             | 87.55| 77.87   | 99.27  | 75.92    |
| SVFT$^P$        | 49.2K             | 86.67    | 77.47   | 99.09      |  73.52   |
| SVFT$^R_{d=4}$    | 441.5K            | 87.05    | 78.95| 99.23     | 78.9     |
| SVFT$^B_{d=4}$    | 441.5K            | 86.95    | 78.85   | 99.24      | 78.93    |
---

**References** \
[1] He, Xu Owen. "Mixture of A Million Experts." arXiv preprint arXiv:2407.04153 (2024)\
[2] Dawid Jan Kopiczko et al. VeRA: Vector-based Random Matrix Adaptation. ICLR 2024

---

### Comment · Area_Chair_YgWR · 2024-08-07
**Reviewer-Author Discussion**

Dear Reviewers,

Thank you very much for your big efforts.

Now, the authors' rebuttal are available, please check them, as well as other reviewers' comments, to consolidate your comments if needed, and to interact with authors and/or other reviewers and/or ACs for any further clarifications as you see fit.

We appreciate you for the continuous support.

Thank you.

---

> ### Comment · Area_Chair_YgWR · 2024-08-13
> **Let's wrap up the reviewer-author discussion**
>
> Dear Reviewers and Authors,
>
> Thank you very much for all the informative discussions. We appreciate you for the big efforts.
>
> To reviewers: if you need any further clarifications from authors, please make sure they can be addressed by text-only response without additional experimental results.
>
> To authors: If you have any clarification and summary you would like to share with reviewers, or request their comments on some of your rebuttal and/or discussion points, please add those and/or kindly remind the reviewer(s).
>
> Thank you.

---

### Author Response · Authors · 2024-08-13
**Rebuttal Summary**

We sincerely thank the reviewers and area chair for their insightful discussions and feedback. \
\
We are encouraged by their recognition of our work as novel, well-motivated, and interesting (**Cyjn, DEin, BEJi**), as well as easily reproducible (**x5z5**). Reviewers **Cyjn** and **BEJi** particularly valued our extensive experiments across vision and language tasks, while both **DEin** and **Cyjn** highlighted SVFT's comparable or superior performance to existing PEFT methods. **DEin** also observed that SVFT's increase in trainable parameters is lower compared to LoRA/DoRA, allowing for greater adaptability across layers. Furthermore, **DEin** noted that SVFT is more effective at reducing trainable parameters than VeRA.

Below is a summary of our responses:

**Discussion on Related Work:** We appreciate the reviewers bringing SVDiff, SVF, and SAM-Parser to our attention. These methods are equivalent to SVFT$^{P}$, and we will acknowledge them in our final version with a detailed discussion. During the rebuttal, we compared these methods against SVFT$^{P}$, which, as expected, showed similar performance (Global Response - **G1**). \
\
We re-emphasize that our primary contribution lies in making *additional* off-diagonal elements in $M$ learnable, facilitating a smooth trade-off between trainable parameters and expressivity—an aspect that significantly differentiates our approach from previous works leveraging singular values.\
\
**Memory Consumption:** We summarized the peak GPU memory consumption of SVFT, LoRA, and DoRA (Global Response - **G2**). Our results show that for an equivalent number of trainable parameters, SVFT consumes comparable or less memory than DoRA, while achieving higher accuracy than both DoRA and LoRA.\
\
**Choice of Sparsity Pattern:** We conducted additional ablation studies across multiple models (Gemma-2B, Gemma-7B, Llama-3-8B) to explore all sparsity patterns evaluated in this work (**x5z5 - W2/Q3**). SVFT$^B$ consistently provided more robust gains, while other variants demonstrated competitive performance. Designing task-specific patterns, however, is beyond the scope of this work.\
\
**Inconsistent Evaluation:** We re-ran experiments with SVFT adapting only QKVUD target modules, ensuring consistency with the baselines (LoRA and DoRA). Even under this consistent setting, SVFT maintained its performance gains.\
\
As the rebuttal phase draws to a close, we welcome any further questions or clarifications from the reviewers. We believe our extensive experiments and detailed responses have addressed their concerns, and we kindly request their favorable consideration of our submission.

---

### Decision · Program_Chairs · 2024-09-25

**Decision:**

Accept (poster)

**Comment:**

This paper presents a parameter-efficient fine-tuning (PEFT) method based on singular value decomposition (SVD) which highlights the importance of learning off-diagonal coefficients. The majority of consensus from the reviewers was that the effectiveness of finetuning off-diagonal coefficients provides a new sight in the literature of PEFT.  Reviewers (DEin, BEJi and Cyjn) are positive, while Reviewer x5z5 recommended to reject it. During the reviewer-author discussion period, there are new experimental results which seem to show the advantages of the proposed method, as confirmed by reviewers.  During the reviewer-AC discussion, there are informative discussions between reviewers in term of the novelty of the proposed method. The key discussion point is whether it is possible, and how to, design the sparsity pattern for the off-diagonal coefficients for downstream tasks in a more elegant way to go beyond the proposed three heuristic methods in the submission. Both the reviewers and the meta-reviewer agree it is an important aspect.  Overall, this meta-reviewer concurs with the overall positive feedbacks in terms of the effectiveness of the proposed method, and recommends to **accept** this paper.

The authors are encouraged to carefully revise the paper due to **the significant updates in the rebuttal and discussion**. The authors are also encouraged to discuss the design and/or the learning of the sparsity pattern in the revision.